# Sox17 and β-catenin co-occupy Wnt-responsive enhancers to govern the endoderm gene regulatory network

**Shreyasi Mukherjee[1,2†], Praneet Chaturvedi[1,2†], Scott A Rankin[1,2†], Margaret B Fish[3], Marcin Wlizla[1‡], Kitt D Paraiso[3,4], Melissa MacDonald[1,2], Xiaoting Chen[5], Matthew T Weirauch[2,5], Ira L Blitz[3], Ken WY Cho[3], Aaron M Zorn[1,2]***

[1]Center for Stem Cell and Organoid Medicine (CuSTOM), Division of Developmental Biology, Perinatal Institute, Cincinnati Children's Hospital Medical Center, Cincinnati, United States; [2]University of Cincinnati, College of Medicine, Department of Pediatrics, Cincinnati, United States; [3]Department of Developmental and Cell Biology, University of California, Irvine, Irvine, United States; [4]Center for Complex Biological Systems, University of California, Irvine, Irvine, United States; [5]Center for Autoimmune Genomics and Etiology (CAGE), Divisions of Biomedical Informatics and Developmental Biology, Cincinnati Children's Hospital Medical Center, Cincinnati, United States

**\*For correspondence:**
aaron.zorn@cchmc.org

[†]These authors contributed equally to this work

**Present address:** [‡]The National *Xenopus* Resource, Marine Biological Laboratory, Woods Hole, United States

**Competing interests:** The authors declare that no competing interests exist.

**Abstract** Lineage specification is governed by gene regulatory networks (GRNs) that integrate the activity of signaling effectors and transcription factors (TFs) on enhancers. Sox17 is a key transcriptional regulator of definitive endoderm development, and yet, its genomic targets remain largely uncharacterized. Here, using genomic approaches and epistasis experiments, we define the Sox17-governed endoderm GRN in *Xenopus* gastrulae. We show that Sox17 functionally interacts with the canonical Wnt pathway to specify and pattern the endoderm while repressing alternative mesectoderm fates. Sox17 and β-catenin co-occupy hundreds of key enhancers. In some cases, Sox17 and β-catenin synergistically activate transcription apparently independent of Tcfs, whereas on other enhancers, Sox17 represses β-catenin/Tcf-mediated transcription to spatially restrict gene expression domains. Our findings establish Sox17 as a tissue-specific modifier of Wnt responses and point to a novel paradigm where genomic specificity of Wnt/β-catenin transcription is determined through functional interactions between lineage-specific Sox TFs and β-catenin/Tcf transcriptional complexes. Given the ubiquitous nature of Sox TFs and Wnt signaling, this mechanism has important implications across a diverse range of developmental and disease contexts.

## Introduction

During embryogenesis, the pluripotent zygote progressively gives rise to specialized cell types expressing distinct sets of genes that, in turn, define the cell's identity and encode proteins necessary for its function. Lineage-specific gene expression is controlled by the genomic integration of signaling pathways and transcription factors (TFs) on DNA cis-regulatory modules (CRMs), such as enhancers, that control transcription (*Heinz et al., 2015*; *Stevens et al., 2017*). An important goal of developmental biology is to elucidate how signaling effectors and TFs interact on distinct sets of CRMs within the chromatin landscape to form gene regulatory networks (GRNs) that activate lineage-specific transcriptional programs whilst repressing expression of alternative fates

(*Charney et al., 2017b*). This will provide a deeper understanding of how transcriptional networks are established, dysregulated in disease, and how GRNs might be manipulated for therapeutic purposes.

We have addressed this in the context of endoderm germ layer specification: one of the earliest cell fate decisions in vertebrate development that provides a relatively simple model to elucidate how GRNs control lineage-specific transcriptional programs (*Charney et al., 2017b*). In embryos and pluripotent stem cells (PSCs), Nodal/Smad2 and Wnt/ß-catenin (Ctnnb1; hereafter Bcat) signaling cooperate to initiate the mesendoderm program and subsequent development of definitive endoderm progenitors, which gives rise to the epithelia of the digestive and respiratory systems (*Sumi et al., 2008*; *Zorn and Wells, 2009*). Downstream of Nodal and Wnt, a core set of endoderm TFs: Sox17, Gata4-6, Eomes, Foxa1/2 and Mix family of homeodomain TFs execute the endoderm GRN (*Arnold et al., 2008*; *Charney et al., 2017b*; *Engert et al., 2013*; *Sinner et al., 2006*). In *Xenopus* where endoderm specification is well studied, the Nodal and Wnt pathways interact at several levels. First, maternal (m) Wnt/Bcat, active on the dorsal side of the blastula, is required for high levels of *nodal* expression at the onset of zygotic transcription (*Hyde and Old, 2000*). Then Nodal and mWnt cooperate to promote the expression of endoderm TFs Sox17, Foxa2 and many dorsal mesendoderm organizer genes (*Xanthos et al., 2002*). A few hours later, zygotic (z) *wnt8* (a Nodal target) is expressed on the ventral side of the gastrula where, together with Nodal signaling, it promotes ventral and posterior mesendoderm identities (*Charney et al., 2017b*; *Stevens et al., 2017*). In mammals, early Wnt/Bcat similarly activates expression of Nodal and core endoderm TFs, with prolonged Wnt promoting posterior hindgut fate (*Engert et al., 2013*; *Zorn and Wells, 2009*).

Functional interactions between the core endoderm TFs and the Wnt pathway are thought to (1) segregate the transient mesendoderm into endoderm and mesoderm, (2) pattern the nascent endoderm into spatially distinct subtypes and (3) execute the downstream differentiation program to give rise to endoderm-derived lineages. But mechanistically how the Wnt signaling machinery interacts with core endoderm TFs to execute this differentiation cascade is unresolved. The transcriptional targets of the endoderm TFs are largely unknown, and it is unclear exactly how Wnt/Bcat regulates distinct spatiotemporal transcription programs in the embryo. Indeed, how the canonical Wnt pathway elicits context-specific transcriptional responses in its multitude of different biological roles in development, homeostasis and cancer is still poorly understood. According to the dogma of canonical Wnt signaling, Wnt-activated Fzd-Lrp5/6 receptor complexes sequester the Bcat degradation complex, resulting in the stabilization and translocation of Bcat to the nucleus. There, it interacts with one of four HMG-box Tcf TFs (Tcf7, Tcf7L1, Tcf7L2 and Lef1) at Wnt-responsive CRMs (*Cadigan and Waterman, 2012*; *Schuijers et al., 2014*), ultimately activating a multiprotein 'Wnt-enhanceosome' with the scaffold proteins Bcl9, Pygopus and the ChiLS complex (*Gammons and Bienz, 2018*). In the absence of Bcat, Tcfs are coupled with corepressor proteins Tle/Groucho and histone deacetylase Hdac to inhibit transcription of Wnt target genes. Bcat displaces Tle and recruits a co-activator complex including histone acetyltransferases Ep300 or CBP to stimulate transcription (*Cadigan and Waterman, 2012*). How distinct context-specific Wnt target genes are selected is unclear since all Tcfs have nearly identical DNA-binding specificities (*Badis et al., 2009*; *Ramakrishnan and Cadigan, 2017*) and for the most part they are ubiquitously expressed. An emerging idea is that the Wnt-enhanceosome also interacts with other lineage-specific TFs to integrate lineage-specific inputs (*Gammons and Bienz, 2018*; *Trompouki et al., 2011*); yet, how these impact genomic specificity in vivo is largely untested and the idea of Tcf-independent Wnt-mediated transcription remains controversial.

In this study, we investigated the possibility that Sox17 functionally interacts with Wnt/Bcat to regulate transcription in the *Xenopus* endodermal GRN. In all vertebrate embryos, Sox17 is specifically expressed in the gastrula endoderm where it is required for early gut development (*Clements et al., 2003*; *Hudson et al., 1997*; *Kanai-Azuma et al., 2002*; *Viotti et al., 2014*). Despite the critical role of Sox17 in endoderm development, only few of its direct transcriptional targets have been identified (e.g.: *hnf1b*, *foxa1* and *dhh*) (*Ahmed et al., 2004*; *Sinner et al., 2004*; *Yagi et al., 2008*). In *Xenopus*, ectopic Sox17 is sufficient to initiate endoderm development in pluripotent blastula animal cap cells (*Clements et al., 2003*; *Hudson et al., 1997*), and co-injection of stabilized Bcat can enhance this activity (*Sinner et al., 2004*). Sox17 can physically interact with Bcat in vitro and suppress the transcriptional activity of generic Tcf/Bcat reporter constructs (TOPflash) in tissue culture experiments (*Sinner et al., 2007*; *Zorn et al., 1999*). However, the biological relevance

of these interactions and whether Sox17 and Bcat functionally interact on chromatin to regulate the endoderm GRN remains unknown.

Here, we define the genomic targets of Sox17 in the *Xenopus* gastrula. In addition to promoting expression of endoderm genes, Sox17 also represses ectoderm and mesoderm gene transcription, and acts in a negative feedback loop to restrain Nodal signaling. We demonstrate that functional interactions with canonical Wnt signaling is a key feature of the Sox17-regulated GRN. Over a third of all Bcat and Sox17 genomic binding in the gastrula occur at the same CRMs. In some instances, Sox17 suppresses Bcat-Tcf mediated transcription, while in other cases, Sox17 and Bcat synergistically activates enhancers apparently independently of Tcfs, indicating a novel mode of regulation. These results provide new insight into the GRN controlling endoderm development and have implications for how Sox TFs and Bcat might interact in diverse biological contexts from development to cancer.

## Results

### Sox17 regulates a genomic program controlling germ layer segregation and endoderm development

To identify the transcriptional program regulated by Sox17, we performed RNA-sequencing (RNA-Seq) on control and Sox17-depleted *Xenopus tropicalis* embryos at multiple time points during blastula and gastrula stages (NF9-12) when the endoderm germ layer is being specified. In *Xenopus tropicalis*, there are three redundant genes: *sox17a*, *sox17b.1* and *sox17b.2* (collectively *sox17*) with indistinguishable activities and identical expression in presumptive vegetal endoderm cells of gastrula embryos (*Figure 1A* and *Figure 1—figure supplement 1A*; *D'Souza et al., 2003*; *Hellsten et al., 2010*). Microinjection of a combination of antisense morpholino oligos (sox17aMO and sox17bMO) targeting all three paralogs resulted in a robust knockdown of Sox17 protein as confirmed by immunostaining (*Figure 1B* and *Figure 1—figure supplement 1B*). The Sox17-MO phenotype was consistent with previous reports (*Clements et al., 2003*; *Sinner et al., 2006*) and phenocopied mouse mutants with defective gut development (*Kanai-Azuma et al., 2002*). Injection of mRNA encoding mouse Sox17 rescued both the anatomical and gene expression phenotypes confirming the efficacy and specificity of the MOs (*Figure 1F* and *Figure 1—figure supplement 1C, D*).

Differential expression analysis of control-MO and Sox17-MO embryos identified 1023 Sox17-regulated genes ($\geq$2 fold change, FDR< 5%) (*Figure 1D* and *Supplementary file 1*). Gene Ontology (GO) enrichment was consistent with the Sox17-regulated transcriptome being involved in 'endoderm formation', 'epithelial differentiation' and 'digestive track morphogenesis' (*Figure 1—figure supplement 1E*). In total, 493 transcripts were downregulated in Sox17-depleted embryos and 530 transcript were upregulated. The time course data revealed that >75% of the differentially expressed genes were changed by Sox17 depletion (either directly or indirectly) during the mid-late gastrula (NF10.5–12), consistent with a role in maintaining endodermal fate after initial induction by Nodal signaling in the blastula (NF9) (*Figure 1D* and *Figure 1—figure supplement 1A*). Differentially expressed genes include 73 TFs, including known Sox17 targets *foxa2* and *hnf1b* (*Sinner et al., 2004*; *Sinner et al., 2006*), and paracrine signaling components (*Supplementary file 1*) such as the Hedgehog pathway (*dhh*, *hhip*, *gli1*); a key epithelial signal in gut organogenesis. This confirms that Sox17 sits atop of the regulatory hierarchy regulating endoderm differentiation.

We next investigated how Sox17-regulated genes were spatially expressed, leveraging a previously published RNA-seq of different tissues dissected from gastrula embryos (*Blitz et al., 2017*). As predicted, a majority of Sox17-dependent genes were enriched in the vegetal endoderm and dorsal mesendoderm (organizer) (*Figure 1E*) including *osr1*, *hnf1b*, *dhh* and *slc5a8* (*Figure 1F* and *Figure 1—figure supplement 1D*). Interestingly, over 30% of the genes upregulated early (NF9-10) in the Sox17-depleted embryos were normally enriched in the ectoderm or mesoderm tissue; examples include ectoderm-promoting TFs *tfap2a* and *lhx5* (*Houston and Wylie, 2003*; *Figure 1D–F*) as well as the mesoderm TFs *foxf1*, *tbx20* and *hlx*. This suggests that Sox17 plays an important role in repressing ectoderm and mesoderm fate in vegetal endoderm cells. Unexpectedly Sox17 also negatively regulated ~150 genes that are normally enriched in the vegetal endoderm and dorsal mesendoderm. Some of these vegetally enriched genes encoded components of the endoderm promoting

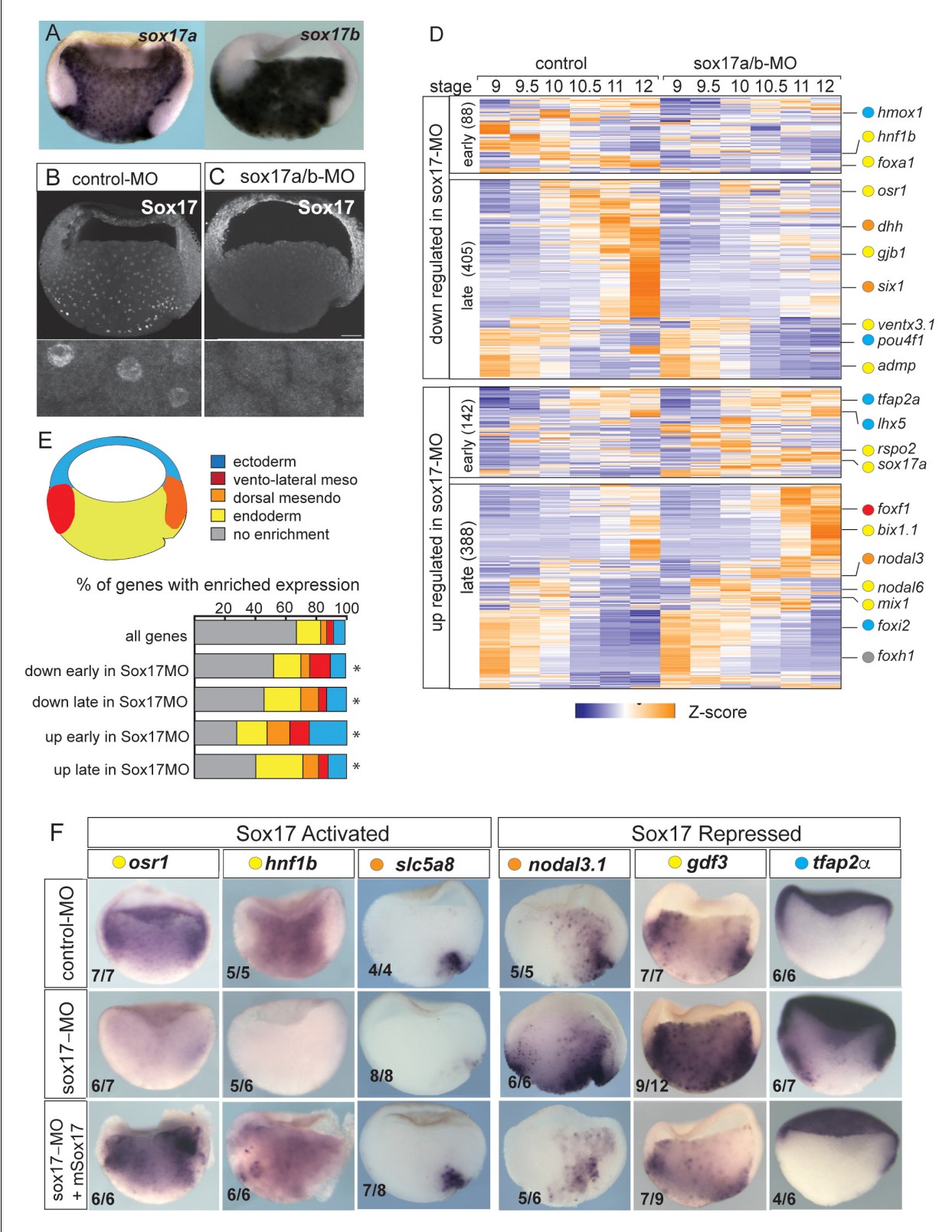

**Figure 1.** Sox17-regulated endoderm transcriptome. (**A**) Endoderm expression of *sox17a* and *sox17b* in NF10.5 *Xenopus tropicalis* gastrula. (**B–C**) Sox17 immunostaining in control-MO (**B**) and sox17a/b-MO (**C**) injected gastrula with an antibody that recognizes both Sox17a and Sox17b shows MO effective knockdown. (**D**) Time course heatmap of Sox17-regulated transcripts. Differentially expressed genes from a pairwise comparison of control-MO and sox17-MO (≥2 fold change, FDR < 5%) at each stage showing transcripts that are downregulated or upregulated early (NF9-10) or late

*Figure 1 continued on next page*

*Figure 1 continued*

(NF10.5–12). Key genes are color coded based on regional expression from fate map in (E). (E) Enriched expression of Sox17-regulated transcripts. Gastrula fate map (dorsal right) colored by different tissues. Stacked bar graphs showing different patterns of enriched spatial expression based on RNA-seq of dissected gastrula tissues (*Blitz et al., 2017*). Distribution of expression patterns statistically different from all genes in the genome was determined by two-sided Kolmogorov-Smirnov tests *p<0.01. (F) In situ validation of Sox17-regulated transcripts. Disrupted expression in sox17-MOs is rescued by co-injection of mouse (m) *Sox17* RNA.

The online version of this article includes the following figure supplement(s) for figure 1:

**Figure supplement 1.** Characterization of Sox17-MO embryos and Sox17-regulated transcriptome.

Nodal pathway including *nodal1, nodal6, gdf3, gdf6, foxh1, mix1* and *bix1.1*, all of which were upregulated in Sox17-depleted embryos (*Figure 1D–F* and *Figure 1—figure supplement 1*). Indeed, even *sox17* transcripts were modestly increased in Sox17-MO embryos even though previous work has demonstrated that Sox17 directly maintains its own transcription (*Howard et al., 2007*) after initial induction by Nodal/Smad2. These data indicate that Sox17 acts as a negative feedback inhibitor that restricts excessive endoderm development by restraining Nodal activity after initial induction.

The observation that over 10% of Sox17-regulated genes are enriched in the organizer mesendoderm while Sox17 is present throughout the endoderm, suggests that Sox17 might also functionally interact with the Wnt and/or Bmp dorso-ventral patterning pathways to control spatial expression. Intriguingly, we found that Sox17 regulates the expression of several Wnt/Bcat pathway components and targets including: *dkk1, dkk2, fzd5, nodal3.1, pygo1, sia1, ror1, rsop2* and *wnt11* (*Figure 1D,F* and *Supplementary file 1*). Together these data suggest that Sox17 regulates both endoderm specification and patterning while suppressing mesectoderm fate and that Sox17 participates in feedback loops with the Wnt and Nodal pathways.

## Sox17 ChIP-Seq reveals direct endodermal targets and a Nodal and Wnt feedback

To identify direct Sox17 targets we generated and validated anti-Sox17 antibodies (*Figure 2—figure supplement 1A–C*) and performed Sox17 ChIP-seq in gastrula (NF10.5) embryos, identifying 8436 statistically significant Sox17-bound putative CRMs (IDR; p<0.05) (*Supplementary file 2*). These were associated with 4801 genes (*Figure 2A*), based on annotation to the nearest transcription start-site (TSS) by HOMER (*Heinz et al., 2010*). 88% of Sox17-bound loci were in introns or intergenic regions more than 1 kb away from the TSS, consistent with Sox17 binding at distal CRMs (*Figure 2—figure supplement 2*). A comparison to published ChIP-seq data of *Xenopus* embryos at the same stage (*Hontelez et al., 2015*) showed that most of the Sox17-bound genomic loci were also bound by Ep300, (*Figure 2B,E*) indicative of active enhancers. Motif analysis of the ChIP-seq peaks confirmed that Sox17 motifs were the most enriched, as expected (*Figure 2D* and *Figure 2—figure supplement 2*). Control ChIP-qPCR experiments of Sox17-MO embryos showed binding reduced to near background levels at 9 of 10 loci tested confirming a robust knockdown and the specificity of the Sox17 antibody (*Figure 2—figure supplement 1D*).

A comparison to published human ChIP-Seq data revealed that 20% of the *Xenopus* Sox17-bound genes were also SOX17-bound in hPSC-induced definitive endoderm (*Figure 2—figure supplement 2D*; *Tsankov et al., 2015*). GO analysis of the conserved *SOX17*-bound genes show an enrichment for 'Tgfb receptor activity' and 'Bcat binding' (*Figure 2—figure supplement 2E*), reinforcing the notion that functional interaction with the Nodal and Wnt pathways is a conserved feature of the Sox17-regulated endoderm GRN.

Intersecting the RNA-seq and ChIP-seq data identified 315 genes associated with 609 Sox17-bound enhancers, which are likely to be direct transcriptional targets (*Figure 2A*). These putative direct targets had significantly enriched expression in the endoderm or dorsal mesendoderm (44%, 139/315 transcripts) (*Figure 2C*). Of the Sox17-bound and regulated genes, 197 genes (383 peaks) were positively regulated by Sox17 (down in Sox17MOs) including *slc5a8, hnf1b, foxa1* and *dhh* (*Figure 2E*), all of which were previously suggested to be direct Sox17-targets. Sox17 negatively regulated 118 putative direct targets (up in Sox17MOs) including ectodermal genes (*lhx5, foxi2* and *tfap2a*), endoderm-enriched Nodal pathway genes (*nodal1, gdf3, gdf6* and *mix1*) and Wnt-regulated organizer genes (*dkk* and *fst*), suggesting direct transcriptional repression (*Figure 2E*).

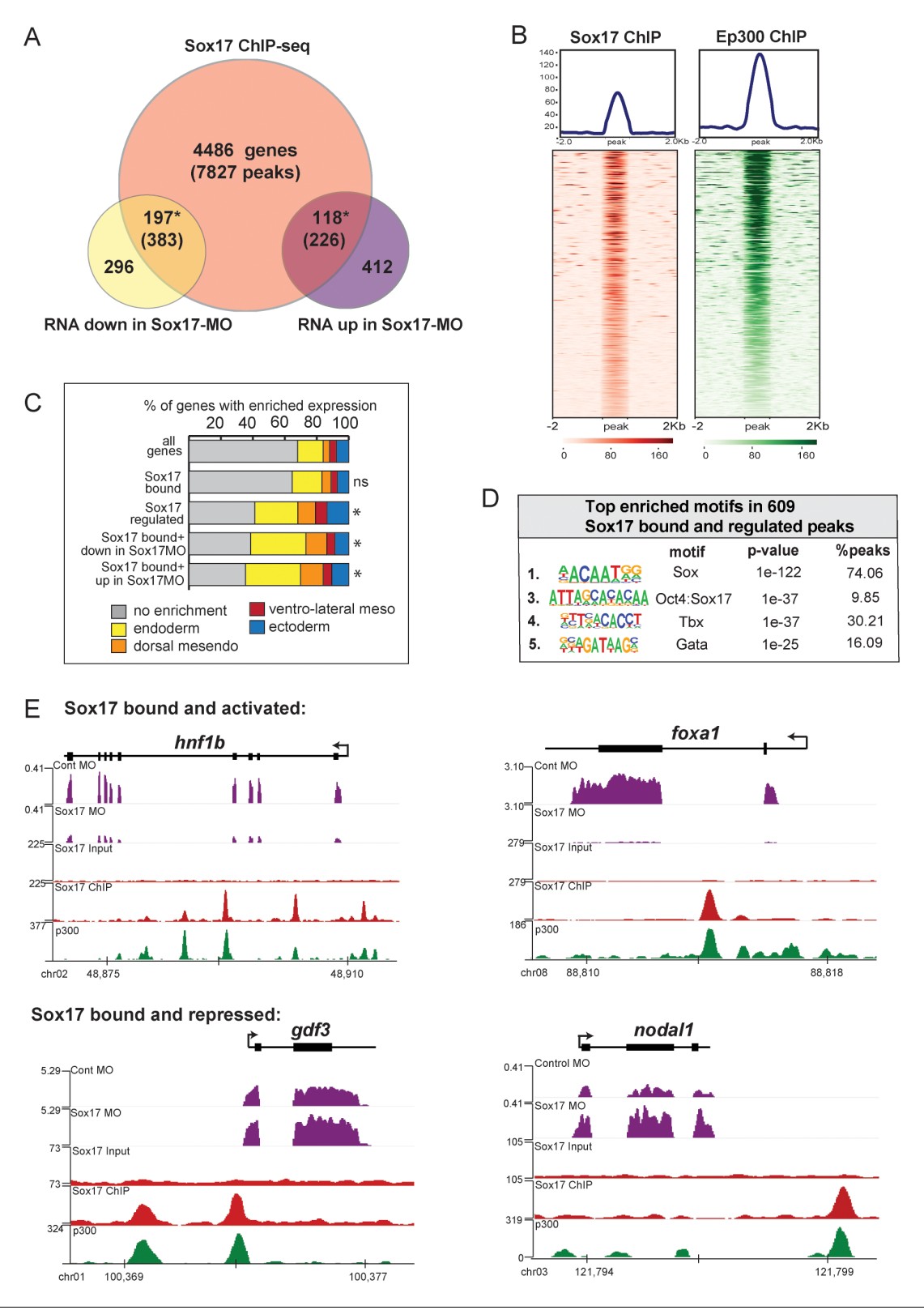

**Figure 2.** ChIP-seq identifies Sox17-bound enhancers and direct transcriptional targets. Sox17 ChIP-seq of gastrula embryos. (A) Venn diagram of Sox17-bound genes (and associated peaks) from ChIP-seq intersected with Sox17-regulated genes. Gene intersections are statistically significant based on hypergeometric tests* p<0.001. (B) Peak density plots showing that the Sox17-bound loci are cobound by the histone acetyltransferase Ep300, a marker of active enhancers. Average density of all peaks in top panel. (C) Sox17-bound genes are enriched in the endoderm. Stacked bar graphs

*Figure 2 continued on next page*

*Figure 2 continued*

showing different patterns of enriched spatial expression based on RNA-seq of dissected gastrula tissues (*Blitz et al., 2017*). Distribution of expression patterns statistically different from all genes in the genome was determined by two-sided KolmogorovSmirnov tests *p<0.01. (D) Top enriched DNA-binding motifs in Sox17 peaks, based on HOMER hypergeometric test. (E) Genome browser views of representative genes showing RNA-seq expression in control-MO and sox17-MO embryos and ChIPseq tracks of Sox17 and Ep300 binding.

The online version of this article includes the following figure supplement(s) for figure 2:

**Figure supplement 1.** Validation of ChIP antibodies.

**Figure supplement 2.** Analysis of Sox17-bound chromatin loci.

Interestingly, ~45% of peaks associated with Sox17-activated genes were also enriched for LIM-homeodomain binding sites, in contrast to peaks from Sox17-repressed genes which were enriched for Tbx or Pou motifs (*Figure 2—figure supplement 2B*). This suggests that Sox17 may coordinately engage enhancers with other core endoderm GRN TFs and mediate activation or repression of target genes depending on the interacting TFs.

These analyses provide new insight into the endoderm GRN and reveal previously unappreciated roles for Sox17 in germ layer segregation and endoderm patterning involving functional interactions with the Nodal and Wnt pathways. These findings, together with our previous work demonstrating that Sox17-Bcat can physically interact in vitro (*Sinner et al., 2007*; *Zorn et al., 1999*) prompted us to their genomic interactions.

## β-catenin directly regulates the endodermal transcriptome

To test the hypothesis that Sox17 and Bcat functionally interact in vivo, we set out to identify Bcat-regulated genes and compare these to the Sox17-regulated transcriptome. Injection of a well-characterized Bcat-MO (*Heasman et al., 2000*) resulted in (a) depletion of nuclear Bcat, (b) reduced expression of a transgenic Wnt-responsive reporter (*Tran et al., 2010*) and (c) the expected ventralized phenotype, which was rescued by co-injection of stabilized human Bcat mRNA (*Figure 3—figure supplement 1*). Together, this confirms the efficacy and specificity of the Bcat knockdown. RNA-seq of control-MO and Bcat-MO depleted embryos at eight time points from blastula and gastrula stages (NF7-12) identified a total of 2568 Bcat-regulated genes ($\geq$2 fold change, FDR $\leq$ 5%), 1321 of which were downregulated and 1247 upregulated in the Bcat-MO embryos (*Figure 3A* and *Supplementary file 3*). Remarkably, the Bcat-dependent genes encoded 251 TFs, (~20% of all the TFs in the genome), reinforcing the notion that Wnt/Bcat initiates a transcriptional cascade in the early embryo.

Intersecting the Bcat-MO RNA-seq data with previously published Bcat ChIP-seq data from *Xenopus tropicalis* gastrula (*Nakamura et al., 2016*) identified 898 putative direct target genes associated with 2616 Bcat-bound CRMs (*Supplementary file 4*). In the Bcat-MO, 546 genes were downregulated , and 352 genes had increased expression in Bcat-MO embryos (*Figure 3B*). These included almost all the previously known Bcat targets in early *Xenopus* embryos and had extensive overlap with other recent genomic analysis of Wnt targets in the *Xenopus* gastrula (*Ding et al., 2017*; *Kjolby and Harland, 2017*; *Nakamura et al., 2016*; *Figure 3—figure supplement 2D*). Control Bcat ChIP-qPCR of Bcat-MO embryos showed reduced binding to near background levels confirming a robust knockdown (*Figure 2—figure supplement 1E*).

As expected, many of the positively regulated Bcat targets had enriched expression in the dorsal mesoderm (*Figure 3D*) including most known organizer genes such as *cer1*, *chrd*, *dkk1*, *frzb*, *fst* and *gsc*. In contrast, ~30% of Bcat-bound genes with increased expression in the early BcatMO embryos (NF7-9) were enriched for ectoderm-specific transcripts, consistent with the expansion of ectoderm and loss of neural tissue in ventralized embryos (*Figure 3D*). We also identified many direct Bcat targets that are likely to be regulated by zygotic Wnt8, (*Kjolby and Harland, 2017*; *Nakamura et al., 2016*) including *sp5*, *axin2*, *cdx1/2*, *fzd10*, *msgrn1* and eight *hox* genes, all of which were downregulated in Bcat-MOs (*Figure 3A,E*). A number of zygotic Wnt-targets with enriched expression in the ventral mesoderm were components of BMP pathway including; *bambi*, *bmp7*, *id2*, *msx1*, *smad7*, *szl*, *ventx1*, and *ventx3* (collectively known as the BMP synexpression group) (*von Bubnoff et al., 2005*). Most of these genes are known to be directly activated by both Bmp4/Smad1 and zWnt8/Bcat (*Itasaki and Hoppler, 2010*; *Stevens et al., 2017*) but were upregulated, rather than

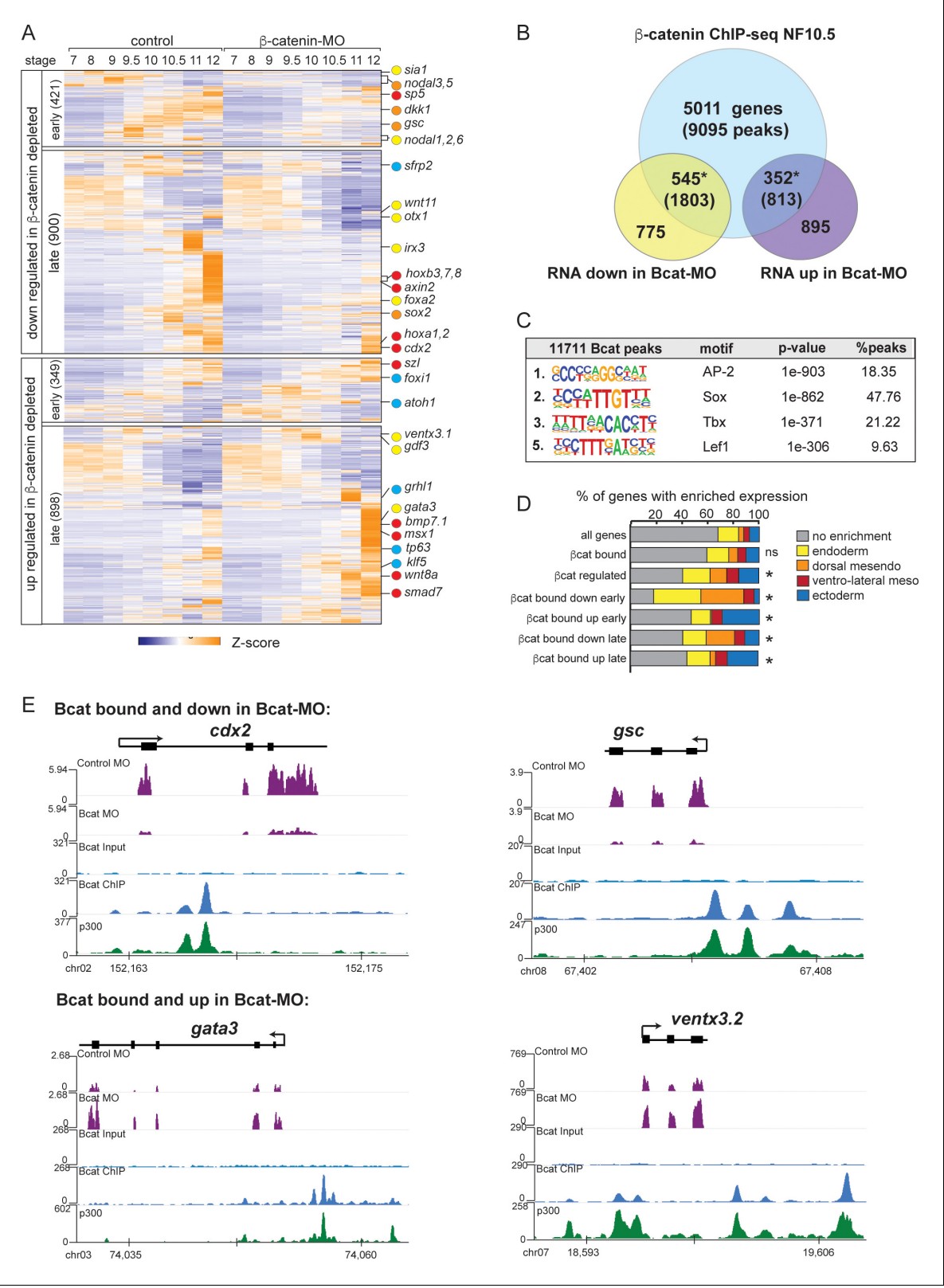

**Figure 3.** The Bcat-regulated genomic program. (**A**) Time course heatmap of Bcat-regulated transcripts. Differential expressed genes from a pairwise comparison of control-MO and Bcat-MO embryos (≥2 FC, FDR < 5%) at each stage identified 2568 Bcat-regulated transcripts; 1321 downregulated and 1247 upregulated in Bcat-MO embryos. Key genes are color coded based on regional expression from (**D**). (**B**) Venn diagram intersecting Bcat-regulated genes with Bcat-bound genes (and associated peaks) from published *X. tropicalis* gastrula Bcat ChIP-seq data (*Nakamura et al., 2016*) *Figure 3 continued on next page*

Figure 3 continued

identified 898 putative direct targets associated with 2616 peaks. Statistically significant intersections based on hypergeometric tests* p<0.001.* (C) Motif enrichment analysis of Bcat peaks by HOMER hypergeometric test. (D) Stacked bar graphs showing different patterns of enriched spatial expression based on RNA-seq of dissected gastrula tissues (*Blitz et al., 2017*). Distribution of expression patterns statistically different from all genes in the genome was determined by twosided Kolmogorov-Smirnov tests *p<0.01. (E) Genome browser views of representative genes showing the RNA-seq expression in control-MO and Bcat-MO embryos and Bcat and Ep300 ChIPseq tracks.

The online version of this article includes the following figure supplement(s) for figure 3:

**Figure supplement 1.** Validation of Bcat-MO embryos.
**Figure supplement 2.** Additional analysis of Bcat ChIP-Seq.

downregulated, in Bcat-depleted embryos, likely due to the increase in BMP/zWnt8 signaling in ventralized embryos.

Most importantly, almost 200 Bcat-bound and -regulated genes had endoderm enriched expression including most of the core endoderm TFs; *sox17*, *foxa*, *eomes*, *gata4*, *mix1* and *mixer* (*Figure 3D* and *Supplementary file 3*). Thus, contrary to the prevailing view that Bcat promotes endoderm fate primarily by activating Nodal ligand expression (*Blythe et al., 2010*; *Hyde and Old, 2000*), we find that Wnt/Bcat also directly regulates the transcription of many endodermal genes. Surprisingly, de novo motif analysis showed that the Bcat-bound peaks were more enriched for Sox than canonical Tcf motifs, 2nd versus 5th ranked, respectively (*Figure 3C*; *Nakamura et al., 2016*). Consistent with this, analysis of other previously published Bcat ChIP-Seq datsets in *X. tropicalis* gastrula embryos (*Gentsch et al., 2019*) and Flag-tagged Bcat ChIP-Seq in *X. laevis* gastrula (*Kjolby and Harland, 2017*) also report a high enrichment for Sox and Tcf motifs (*Figure 3—figure supplement 2A–D*). Together these data support the hypothesis that Sox17 and Bcat coregulate endodermal transcription.

## Sox17 and β-catenin co-occupy endoderm enhancers

Intersection of the ChIP-seq datasets identified 3956 genomic loci co-occupied by both Sox17 and Bcat, and these had epigenetic signatures of active enhancers (*Figure 4A,B*). This represents a third of all Sox17 genomic binding in the gastrula. Comparison of the Sox17-MO and Bcat-MO RNA-seq datasets identified 415 transcripts that were regulated by both Sox17 and Bcat; comprising of ~40% of the Sox17-regulated and ~15% of the Bcat-regulated transcriptome (*Figure 4A*). A similar high overlap between Sox17 and Bcat genomic binding was also observed in another independent Bcat ChIP-seq dataset (*Figure 4—figure supplement 1B*; *Gentsch et al., 2019*). This suggests that co-regulation with Wnt/Bcat is a major feature of the Sox17-regulated endoderm GRN. Analysis of publicly available ChIP-seq revealed that Sox17 and Bcat-bound loci are also enriched for other endodermal TFs including VegT, Foxa4, Smad1 and Smad2 (*Figure 4—figure supplement 1C*; *Charney et al., 2017a*; *Gentsch et al., 2019*), consistent with recent reports that the combinatorial activity of multiple TFs at endodermal super-enhancers (*Paraiso et al., 2019*) coordinate endoderm specification.

In total, 84 genes (191 Peaks) were bound and regulated by both Sox17 and Bcat (*Figure 4A,B*). This is likely to be an underestimation of the number of direct coregulated genes, as these had to pass a stringent thresholding of four independent statistical tests to be included in this list. A comprehensive motif analysis of the 191 co-bound and coregulated peaks using the CIS-BP database (*Lambert et al., 2019*) revealed that 53% (101/191) of the peaks contained both Sox17 and Tcf DNA-binding motifs (e.g. *dkk1*, *lhx5* and *osr1*) compared to 31% in background genomic sequences (*Figure 4—figure supplement 2E* and *Supplementary file 5*). In contrast 24% (46/191) of the peaks had Sox motifs but no Tcf sites (e.g. *six1*), suggesting that Bcat might be recruited to these loci independent of Tcfs. Only 3/191 peaks had Tcf but no Sox motifs. Moreover, co-occupied enhancers often contained multiple Sox17 motifs (>5), which was also significantly enriched over random (*Figure 4—figure supplement 2*). In addition, most individual genes were associated with multiple Sox17 and Bcat co-occupied enhancers, suggesting that combinatorial binding and integration of several CRMs is crucial to control lineage-specific transcription.

Since the ChIP-seq experiments were performed on whole embryos, it was possible that Sox17 and Bcat chromatin binding might occur in different cell populations. To test whether Sox17 and

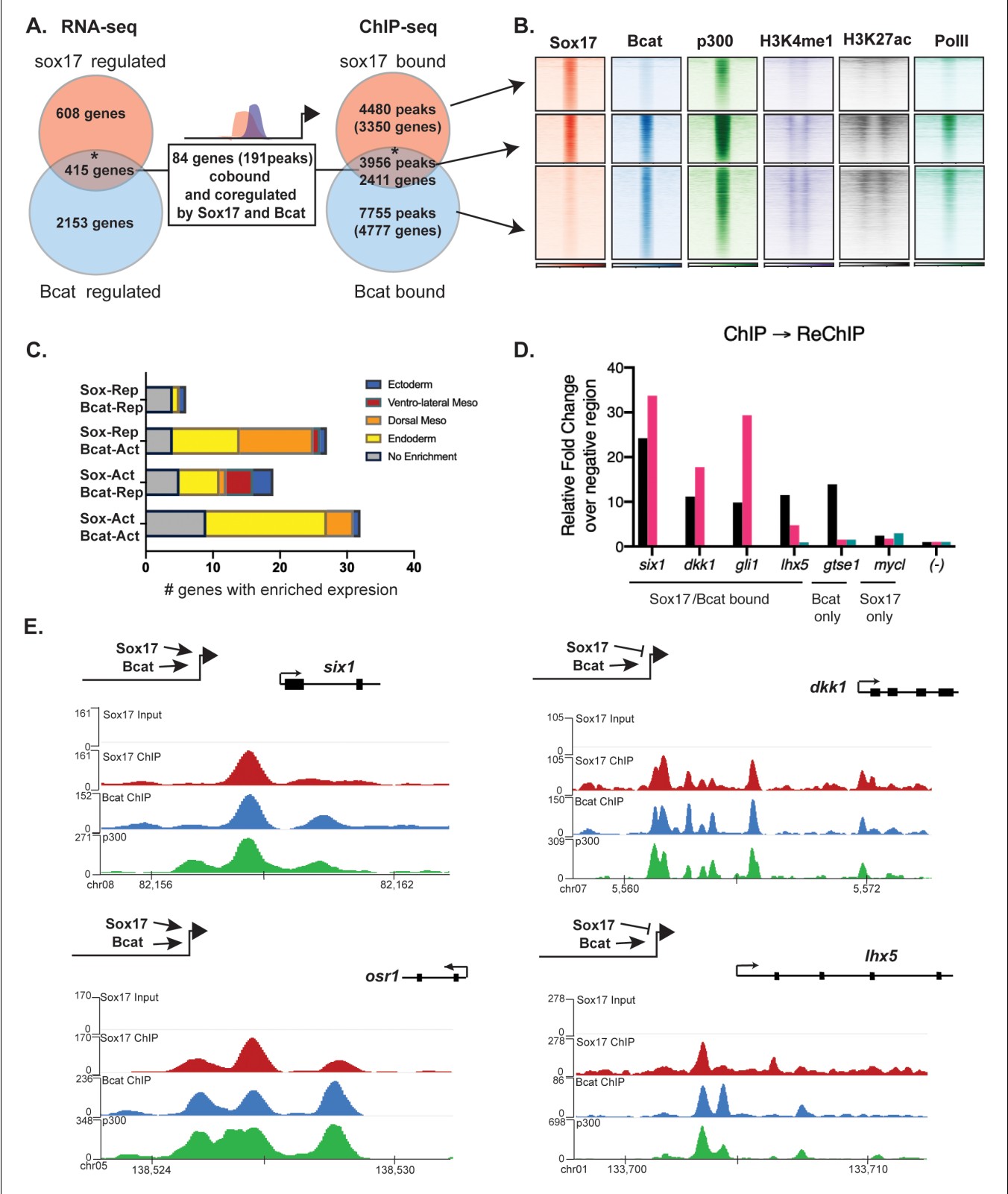

**Figure 4.** Sox17 and Bcat co-occupy enhancers to regulate the endoderm GRN. (**A**) Eighty-four genes (191 Peaks) are cobound and coregulated by Sox17 and Bcat. Venn diagram show that 415 genes are regulated by both Sox17 and Bcat (left), whereas 2411 genes (3956 peaks) are bound by both Sox17 and Bcat. Gene intersections are significant based on hypergeometric tests, *p<0.001. (**B**) Density plots show that Sox17 and Bcat cobound peaks are enriched for enhancer associated epigenetic marks: Ep300, H3K4me1 and H3K27Ac and RNA pol II. (**C**) Regional enrichment of all subsets of
*Figure 4 continued on next page*

*Figure 4 continued*

enhancers cobound and coregulated by Sox17 and Bcat. (D) A representative ChIP-reChIP experiment showing that Sox17 and Bcat coocupy all the test CRMs (*six1*, *dkk1*, *gli1* and *lhx5*) in the same cells, in contrast to the *gtse1* locus, which is bound only by Bcat. (E) Genome browser views of representative genes showing peak overlap of Sox17, Bcat and Ep300 ChIP-Seq tracks.

The online version of this article includes the following figure supplement(s) for figure 4:

**Figure supplement 1.** Additional analysis of cobound and coregulated peaks.

**Figure supplement 2.** Sox and Tcf DNA-binding motif frequency in Sox/Bcat cooccupied peaks.

Bcat bound the same enhancers in the same cells we performed ChIP-reChIP experiments. First anti-Bcat ChIP was performed on 500 gastrulae. DNA eluted from the first ChIP was then re-precipitated with either anti-Sox17 or a negative control IgG antibody. All four loci tested by ChIP-reChIP qPCR (*six1*, *dkk1*, *lhx5* and *gli1*) showed enrichment compared to IgG demonstrating that these genomic loci (~150 bp in length) were simultaneously bound by Sox17 and Bcat in endodermal cells. In contrast, a negative control locus *gtse1* which is bound by Bcat but not Sox17 was not enriched in the ChIP-reChIP experiment (*Figure 4D*).

The 84 co-bound and coregulated genes fell into four regulatory categories (*Figure 4C,E*); 32/84 were activated by both Sox17 and Bcat (e.g. *six1* and *osr1*) and tended to have endoderm enriched expression, 27/84 were activated by Bcat but repressed by Sox17 (e.g. *dkk1*) and these tended to be enriched in the organizer mesendoderm, 19/84 were positively regulated by Sox17 and negatively regulated by Bcat (up in the Bcat MO) and finally 6/84 were negatively regulated by both Sox17 and Bcat. The observation that Bcat positively regulated most of these genes (59/84, 70%) was consistent with its known role in transcriptional activation. Sox17 positively regulated 60% of targets and repressed 40% of targets, suggesting context-dependent activity. For further analysis, we focused on the two major regulatory groups; genes/enhancers activated by both Sox17/Bcat and those activated by Bcat but repressed by Sox17 (*Figure 4E*).

## Sox17 and β-catenin coordinate spatial expression domains in the embryo

It was possible that the overlap in Sox17- and Bcat-regulated genes was largely due to the Sox17-Wnt feedback loop where Bcat is required for robust *sox17* expression and Sox17, in turn, regulates the expression of Wnt-pathway components. While this is likely to be part of the mechanism, we directly tested the functional necessity for both Sox17 and Bcat in epistasis experiments, asking whether injection of *sox17* mRNA could rescue coregulated genes in Bcat-depleted embryos and vice versa. Co-injection of *mSox17* mRNA could not rescue the normal expression patterns of *six1*, *osr1*, *lhx5* or *dkk1* in Bcat-MO embryos, but it did rescue *foxa1* expression (*Figure 5*). This demonstrates that reduced Sox17 in Bcat-depleted embryos cannot account for the disrupted *six1*, *osr1*, *lhx5* or *dkk1* expression, but it can account for the reduced *foxa1* expression. Similarly, co-injection of stabilized (*ca*)Bcat mRNA rescued the normal expression of *foxa1* but not *six1*, *osr1*, *lhx5* or *dkk1* in Sox17-morphants indicating that the defects in their expression were not simply due to reduced Wnt signaling (*Figure 5*).

These experiments also show that Sox17 and Bcat coordinate spatial gene expression in the embryo in a complex gene-specific manner. Wnt/Bcat stimulates expression of *lhx5* in the ectoderm, but Sox17 represses this activity in the presumptive endoderm. Sox17 and Bcat are both required for robust expression of *osr1* and *six1* in the endoderm and dorsal mesendoderm respectively. In contrast, Sox17 normally suppresses Bcat-activated expression of *dkk1* in the deep endoderm. Together, with their co-occupancy at CRMs, these data show that Sox17 and Bcat are both required to directly regulate transcription in discrete spatial domains.

## Sox17 and Bcat synergistically regulate transcription of endodermal enhancers

To examine how Sox17 and Bcat directly regulate endodermal transcription, we first selected four exemplar co-occupied enhancers: (a) a proximal *six1* enhancer located −1 kb from the TSS, (b) a proximal *dkk1* enhancer located −1.6 kb from its TSS, (c) a distal −7.5 kb *dkk1* enhancer and (d) a −1 kb *lhx5* enhancer (*Supplementary file 6*). We focused most of our analysis on *six1* and *dkk1*

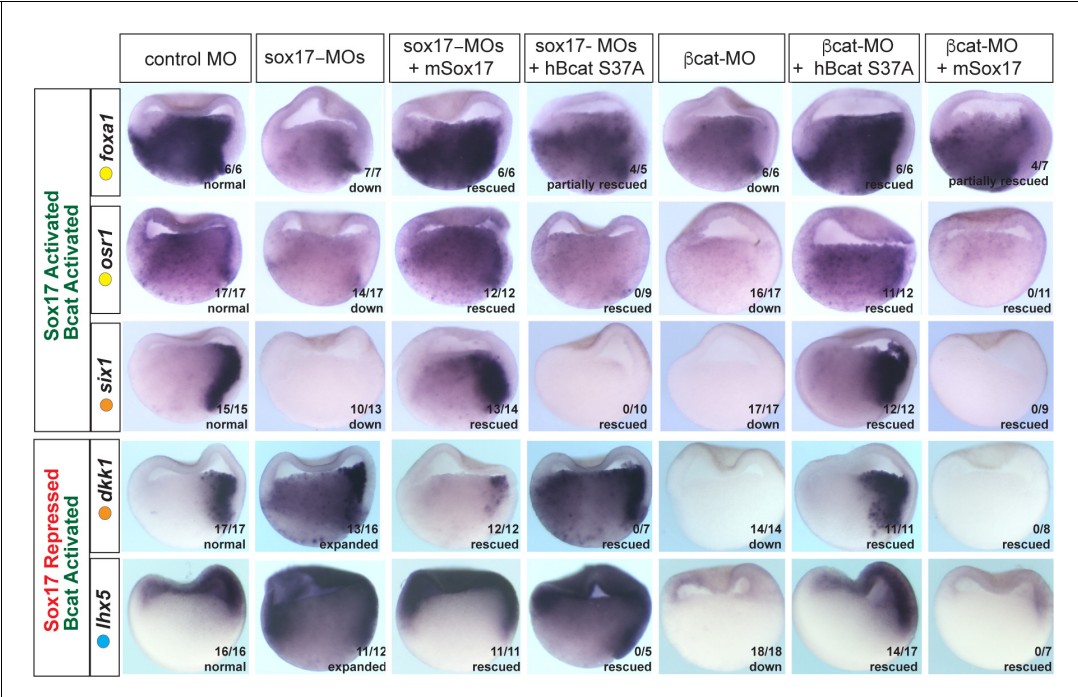

**Figure 5.** Sox17 and Bcat epistasis. In situ hybridization of gastrula embryos (dorsal right) injected with sox17-MO or bcat-MO, some of which were co-injected with RNA encoding either mSox17 or a human S37A stabilized Bcat. The genes *osr1*, *six1* and *foxa1* are activated by both Sox17 and Bcat, whereas *dkk1* and *lhx5* are activated by Bcat but repressed by Sox17. Co-injection of mSox17 cannot rescue normal expression in Bcat-MOs, nor can hBcat S37A RNA rescue Sox17-MOs indicating that both Sox17 and Bcat required for normal expression.

which are both expressed in the dorsal mesendoderm but have opposite regulation. Sox17-Bcat promotes *six1*, whilst Sox17 represses Bcat activation of *dkk1*.

The −1 kb *six1* enhancer is evolutionarily conserved from *Xenopus* to human containing three Sox-binding sites but no predicted Tcf-binding sites (*Figure 6B*). In mouse, this enhancer can drive transgenic expression in the embryonic gut tube, although the TFs that modulate it were previously unknown (*Sato et al., 2012*). Analysis of published ChIP-seq data indicate that, like in *Xenopus*, the human *SIX1* CRM is also bound by SOX17 in human PSC-derived endoderm (*Figure 6—figure supplement 1A*; *Tsankov et al., 2015*). Mammalian *Dkk1* is also known to be a direct Bcat-Tcf target (*Niida et al., 2004*), but the enhancers that regulate its expression in the gastrula had been previously uncharacterized. Unlike the −1 kb *six1* enhancer, the −1.6 kb and −7.5 *dkk1* enhancers as well as the −1 kb *lhx5* enhancer are predicted to harbor both Tcf and Sox DNA-binding sites (*Figure 6B*, *Figure 6—figure supplement 2* and *Supplementary file 6*).

We cloned each of these CRMs into luciferase (luc) reporter constructs and tested whether they require Sox17 and Bcat for enhancer activity in *Xenopus* embryos (*Figure 6A*). Targeted microinjection into different cells of 32-cell-stage *Xenopus* embryos showed that all the reporters recapitulated the endogenous spatial expression indicating that they are *bona fide* enhancers. The *six1* and *dkk1* reporters were both active in the dorsal-anterior endoderm, but not the ventral endoderm or ectoderm, whereas the *lhx5* reporter was only active in the dorsal ectoderm (*Figure 6C,E* and *Figure 6—figure supplement 2*). Depletion of endogenous Sox17 or Bcat abolished expression of the −1*six1*: luc reporter and this was rescued by adding back *mSox17* to Sox17-MO or by adding back *caBcat* mRNA to Bcat-MO embryos (*Figure 6C*). However, co-injection of *mSox17* could not rescue expression in Bcat-MO embryos and nor could *caBcat* RNA rescue expression in Sox17-MOs (*Figure 6C*). These epistasis experiments indicate that both Sox17 and Bcat are required for activation of the −1 kb *six1* enhancer. The human *SIX1:luc* construct exhibited identical activity to the *Xenopus six1:luc* enhancer (*Figure 6—figure supplement 1D*). In contrast, in the case of the −1.6 kb *dkk1*, −7.5 kb *dkk1* and the −1 kb *lhx5:luc* constructs, depletion of Bcat abrogated reporter activity, whereas Sox17 depletion resulted in significantly elevated reporter activity, indicating that Sox17 restricts

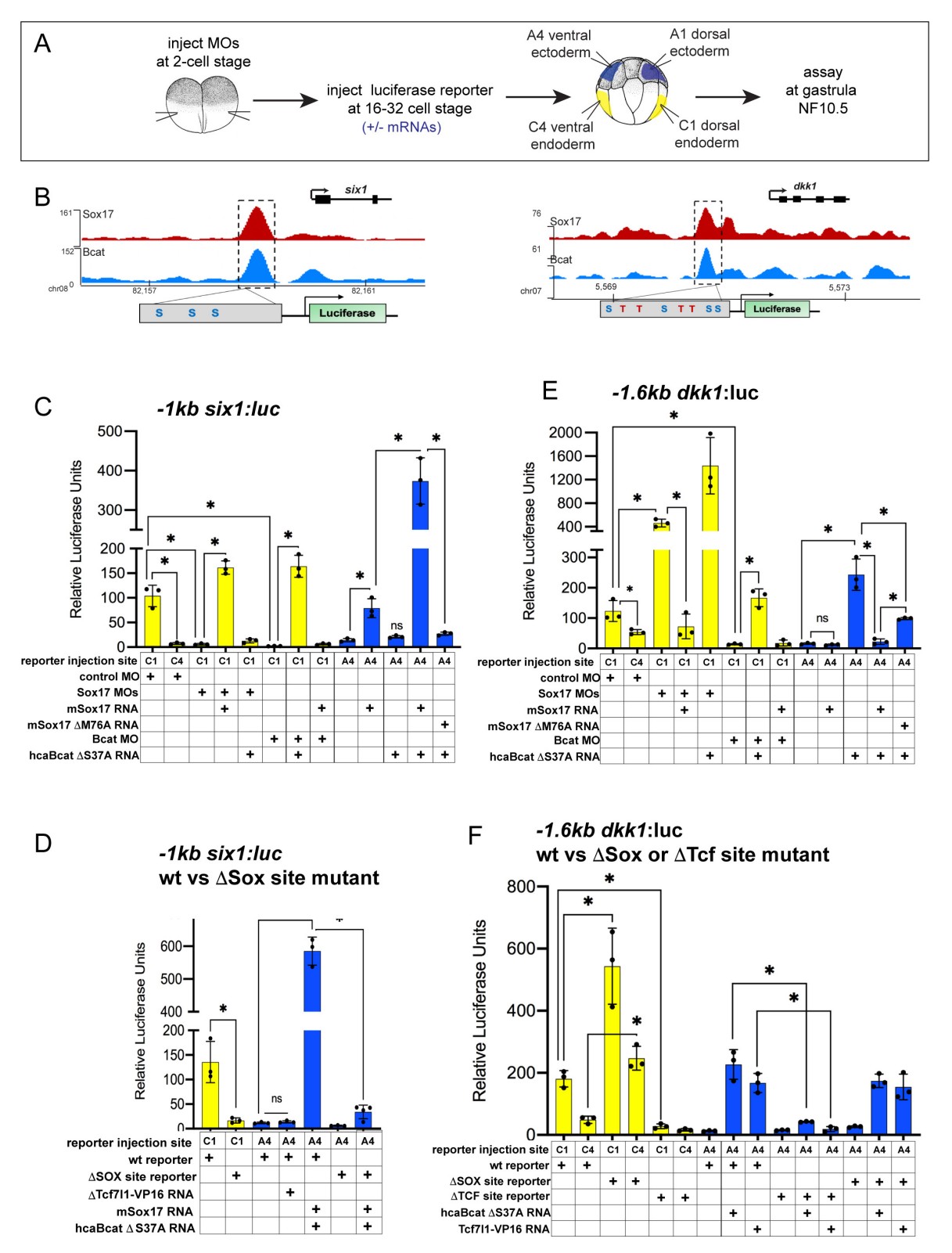

**Figure 6.** Sox17 and Bcat coordinately regulate endoderm enhancers. (A) Experimental design. (B) IGV genome browser tracks showing Sox17 and Bcat chromatin binding and schematic of reporter constructs showing the predicted Sox17 and Tcf DNA-binding sites in the −1 kb *six1* and −1.6 kb *dkk1* enhancers (C–F) Luciferase assay of (C) wildtype (wt) −1 kb *six1:luc*, (D) wt versus DSox-site mutant 1 kb *six1:luc*, (E) wt −1.6 kb *dkk1:luc* and (F) wt

*Figure 6 continued on next page*

*Figure 6 continued*

versus DSox-site and DTcf-site mutant −1.6 kb *dkk1:luc* reporters injected in different tissue with the indicated MOs and/or mRNAs. Histograms show mean luciferase activity ± standard deviation, *p<0.05 in pairwise Student T-tests.

The online version of this article includes the following figure supplement(s) for figure 6:

**Figure supplement 1.** Sox17 and Bcat can regulate some enhancers apparently independent of Tcfs.

**Figure supplement 2.** Reporter assays showing Sox17 and Bcat regulated −7.5 kb *dkk1* and 1 kb *lhx5* enhancers.

Bcat-mediated transactivation at the *dkk1* and *lhx5* enhancers (*Figure 6E*, and *Figure 6—figure supplement 2C,D*).

To test whether Sox17 and Bcat were sufficient to regulate the enhancers, we assayed their activity in ventral ectoderm tissue which does not express endogenous Sox17 and has low Wnt/Bcat activity. Injection of *mSox17* but not *caBcat* mRNA was sufficient to stimulate the −*1 kb six1:luc* and human *SIX1:luc* reporters in the ectoderm, but co-injection of *caBcat* along with *mSox17* synergistically activated the reporters (*Figure 6C* and *Figure 6—figure supplement 1D*). On the other hand, *caBcat* but not *mSox17* was able to activate the *dkk1* and *lhx5* reporters in ventral ectoderm, and co-injection of *mSox17* suppressed the ability of *caBcat* to stimulate these reporters (*Figure 6E* and *Figure 6—figure supplement 2C,D*), consistent with the endogenous regulation (*Figure 5*). The ability of Sox17 to activate the *six1* reporter or suppress Bcat-induced activation of the *dkk1* and *lhx5* reporters required the DNA-binding function of Sox17. A M76A point mutation in mSox17, known to disrupt the DNA-binding HMG domain (*Sinner et al., 2007*) significantly reduced Sox17's ability to activate the *six1* or repress the *dkk1* reporter activity, respectively (*Figure 6C,E* and *Figure 6—figure supplement 2C,D*).

## Sox17 and Bcat appear to stimulate some enhancers independent of Tcfs

The *six1* enhancer is predicted to have only Sox sites with no evidence of Tcf-binding sites, whereas the *dkk1* and *lhx5* enhancers have both Sox17 and Tcf sites. To test whether or not Tcfs regulate these enhancers, we injected a Tcf7l1-VP16 fusion construct which constitutively activates Tcf-target gene transcription (*Darken and Wilson, 2001*). This had little to no impact on the *six1* reporter, but robustly activated the *dkk1* and *lhx5* reporters in ventral ectoderm (*Figure 6D,F* and *Figure 6—figure supplement 2D,E*). We next mutated the Sox17 and Tcf-binding sites in the *six1* and *dkk1* enhancers and examined the impact on Sox17, Bcat and Tcf regulation. Loss of the Sox17 sites abolished normal and Sox17/Bcat-induced expression of the *six1:luc* reporter (*Figure 6D*). In contrast, mutation of the Sox sites in the *dkk1* enhancers resulted in elevated reporter activity consistent with Sox17-mediated repression (*Figure 6F* and *Figure 6—figure supplement 2E*). Deletion of the Tcf sites in the *dkk1* enhancers abolished expression in the dorsal mesendoderm as well as Bcat or Tcf7l1-VP16-induced activation in the ventral ectoderm, consistent with Bcat-Tcf complexes activating these enhancers (*Figure 6F* and *Figure 6—figure supplement 2E*).

To examine the generality of the apparent Tcf-independent regulation of *six1,* we identified two additional enhancers with similar characteristics: an *osr1* enhancer +7 kb downstream of its TSS (*Figure 6—figure supplement 1B*) and a +3 kb *igf3* enhancer (*Figure 6—figure supplement 1B,C* and *Supplementary file 6*). Both these enhancers have no evidence of Tcf binding sites and they were both synergistically activated by Sox17 and Bcat, whereas co-injection of Tcf7l1-VP16 failed to activate either reporter (*Figure 6—figure supplement 1E,F*). From these collective data, we conclude that Sox17 and Bcat synergistically can activate the *six, osr1* and *igf3* enhancers, likely independent of Tcfs. In contrast, the *dkk1 and lhx5* enhancers are activated through a Bcat/Tcf transcription mechanism, with Sox17 spatially restricting Bcat/Tcf activity to maintain cell-type-specific gene expression (*Figure 7*).

## Discussion

### Overview of findings

Sox17 is a key regulator of vertebrate endoderm development and morphogenesis, yet surprisingly, the transcriptional program that it controls has been poorly characterized until now. Here, we

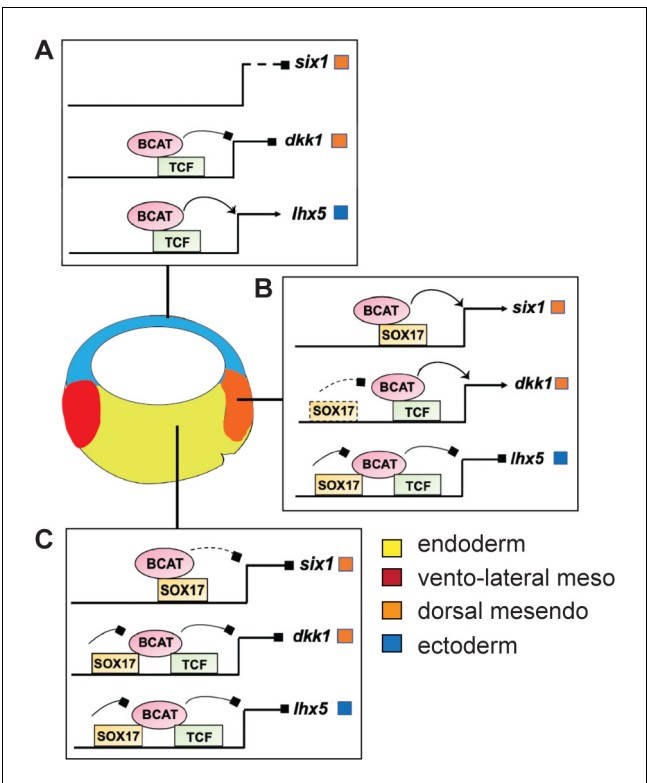

**Figure 7.** Model of Sox17 and Bcat coregulation spatial transcription. Differential engagement of Sox17, Bcat and Tcfs on enhancers in different cells modulates spatial expression domains. (**A**) In ectoderm cells lacking Sox17 (and Nodal signaling), ectoderm-specific gene *lhx5* is activated through a Bcat/Tcf dependent mechanism while *six1* and *dkk1* are not transcribed. (**B-C**) Sox17 is expressed throughout the endoderm and dorsal mesendoderm, while Bcat activity is the higher in dorsal mesendoderm than deep endoderm. (**B**) In the dorsal region, Sox17 and Bcat synergistically coactivate *six1* in the absence of Tcf, whereas Sox17 exerts a repressive influence on Bcat/Tcf-activation of *lhx5 and dkk1*. We postulate, however, that in the dorsal region Sox17-mediated repression is insufficient to overcome Bcat/Tcf activation of *dkk1*. (**C**) In deep endoderm cells where Bcat activity is lower, Sox17-mediated repression of Bcat/Tcf is sufficient to repress *dkk1* and lhx5 transcription, but Bcat activity is not high enough to activate *six1*.

defined the Sox17-regulated GRN in *Xenopus* gastrulae, which includes many conserved *SOX17* genomic targets in PSC-derived human definitive endoderm. We show that in addition to acting atop a transcriptional hierarchy promoting endoderm differentiation, Sox17 has previously unappreciated roles in endoderm patterning and germ layer segregation. We demonstrate that functional interactions with the canonical Wnt signaling pathway is a major feature of the Sox17-regulated GRN. Over a third of all Bcat and Sox17 genomic binding events in the gastrula occur at the same enhancers, where they coordinately regulate spatial gene expression in the embryo. Our data suggest that Sox17 and Bcat synergistically activate a subset of endodermal enhancers apparently independent of Tcfs, whereas at other enhancers Sox17-regulated Bcat-Tcf mediated transcriptional activation (either activating or inhibiting). Together these results provide novel insights into the establishment of the endoderm GRN and suggest a new paradigm where Sox17 acts as a tissue-specific modifier of Wnt-signaling responses.

## The Sox17-Bcat-regulated endodermal GRN

Our genomic study revealed an extensive overlap between Sox17- and Bcat-regulated genes. Together with previous work (*Charney et al., 2017b*; *Zorn and Wells, 2009*), this suggests that Sox17 and the other core endoderm TFs interact with Wnt and Nodal in a series of positive and negative feedback loops to control germ layer segregation and modulate spatial gene expression. Temporally mesendoderm development is initiated by maternal VegT and Bcat, which together initiate

Nodal signaling. We showed that in addition to its known role in activating transcription of endo-derm-inducing *nodal* ligands (*Blythe et al., 2010*; *Hyde and Old, 2000*), Bcat directly promotes the expression of many endodermal genes including most of the core endoderm TFs; *sox17*, *eomes*, *foxa*, *gata4/5*, *mix1* and *mixer*. Sox17 appears to function after initial Nodal signaling in the mid-gastrula stage by promoting expression of endoderm GRN components such as *foxa1* and *gata4*, whilst negatively regulating the expression of some *nodal* and *mix/bix* genes. Thus on one hand, Sox17 promotes endodermal fate, and on the other, it restrains excessive Nodal activity. There is prece-dence for Sox factors influencing *nodal* ligand transcription. Maternal Sox3, localized in the blastula ectoderm, directly binds the *nodal5/6* enhancers to repress Bcat-stimulated transcription, whereas vegetally localized maternal Sox7 enhances *nodal5/6* expression in the vegetal cells (*Zhang and Klymkowsky, 2007*). This suggests a potential of a dynamic handoff model where different Sox TFs might co-occupy the same enhancers with Bcat in different cells to activate or repress transcription in a context-dependent manner. In this model, we predict that in the early blastula Bcat-bound enhancers might be co-occupied by Sox3, Sox7 and/or Tcfs and it is not until the mid-gastrula that co-occupancy of Sox17 and Bcat would be observed.

Our data demonstrate that Sox17-Bcat interactions regulate spatial gene expression in the embryo (*Figure 7*). Sox17-Bcat interact at some enhancers like *six1* and *igf3* to promote endoderm transcription. In addition, Sox17 also has a key role in repressing alternative lineages in the vegetal cells, similar to the maternal TFs Foxh1, Vegt and Otx1 (*Chiu et al., 2014*; *Paraiso et al., 2019*). For example, Sox17 represses Bcat/Tcf-stimulated expression of the ectoderm specifying TF *lhx5* (*Houston and Wylie, 2003*). Sox17-Bcat interactions also impact spatial expression within the endo-derm. Sox17 inhibits the Bcat-Tcf stimulation of the *dkk1* enhancer in the deep endoderm, but in the dorsal mesendoderm where Bcat activity is higher, Sox17 appears insufficient to suppress *dkk1*. The finding that *six1* and *dkk1* are under opposite regulation by Sox17 and Bcat despite having very sim-ilar expression patterns warrants future analysis of the temporal-spatial dynamics in Bcat, Tcf and Sox17 genomic occupancy in dorsal vs ventral cells over time.

## DNA binding specificity of SOX TFs

An important consideration is how Sox17 selects specific target enhancers. In DNA-binding experi-ments, all Sox TFs bind to variants of the DNA sequence 5'-(A/T)(A/T)CAA(A/T)3', which is similar but distinct from Tcf DNA-binding sites 5'-T(A/T)(A/T)CAAG 3' (*Hou et al., 2017*; *She and Yang, 2015*). SOX TFs can act either as monomers or heterodimers with different cell-type specific cofac-tors, which in turn impact their DNA-binding affinity and ability to activate or repress transcription (*Bernard and Harley, 2010*; *Hou et al., 2017*). For example, in PSCs, Oct4-Sox2 heterodimers pro-mote pluripotency and bind to distinct sites than those bound by Oct4-Sox17 complexes that initiate mesendoderm development (*Aksoy et al., 2013*). It is, therefore, likely, that Sox17 and Bcat cooper-ate with other TFs to provide lineage specificity to Wnt-responsive transcription during germ layer segregation. Indeed, motif analysis and comparison to other ChIP-seq datasets revealed that Sox17-bound loci were also enriched for Tbx, Gata, Fox and homeodomain motifs suggesting that in addi-tion to regulating each other's expression, the core endoderm TFs act together in combinatorial manner, perhaps to establish endodermal super enhancers crucial for downstream lineage specifica-tion (*Paraiso et al., 2019*). Interestingly, enhancers that were negatively regulated by Sox17 were enriched for Tbx and Oct motifs whereas enhancers activated by Sox17 were enriched for homeodo-main motifs. This suggests that Sox17 interactions with distinct lineage-specific TFs might modulate context-specific transcriptional responses. This is a particularly attractive hypothesis since as mono-mers Sox TFs bind DNA with low affinity and they are thought to gain DNA-binding specificity by interacting with other TFs (*Hou et al., 2017*). In the future, it will be important to identify and inte-grate the genomic targets of all the core endoderm-specifying TFs and examine how combinatorial enhancer binding is integrated into a unified endodermal GRN.

## Sox-Bcat-Tcf interactions on chromatin

Many Sox proteins, including Sox17, can bind to recombinant Bcat and Tcfs in vitro and modulate Wnt/Bcat-stimulated transcription of synthetic reporters assays with multimerized Tcf-sites (pTOP-Flash) (*Sinner et al., 2007*; *Zorn et al., 1999*). While a number of models have been proposed to explain how Sox factors might impact Bcat activity, (*Kormish et al., 2010*; *She and Yang, 2015*;

*Tan et al., 2019*), in most cases the in vivo biological relevance of these interactions was unclear. Although the resolution of ChIP-reChIP assays only allow us to demonstrate that Sox17 and Bcat bind chromatin within 150 bp of each other, taken together with published in vitro data, our results suggest that Sox-Bcat or Sox-Bcat-Tcf transcriptional complexes can be assembled on enhancers in vivo.

At enhancers like *six1*, *igf3* and +7 kb *osr1*, where Sox17 and Bcat synergistically stimulate transcription, we postulate that Sox17 and Bcat recruit Ep300/Cbp to chromatin, analogous to Tcf-Bcat in canonical Wnt-signaling. Consistent with this, structure-function studies have found that a conserved 10 amino-acid domain (VDXXEFEQYL) in the Sox17 C-terminus is required for both Bcat-binding and transcriptional transactivation (*Sinner et al., 2004*). On the other hand, most Bcat and Sox17 co-occupied enhancers contain multiple Sox and Tcf sites clustered within 50 bp, suggesting that Sox17 and Tcf, together with Bcat and scaffolding proteins like Bcl9, (*Gammons and Bienz, 2018*; *Masuda and Ishitani, 2017*) might assemble higher-order multiprotein complexes facilitating Wnt-mediated transcription. Both Soxs and Tcfs are HMG-domain containing TFs, capable of binding to the minor groove of DNA and inducing DNA bending. This could further indicate that they are brought into close spatial proximity to establish the assembly of functional enhanceosomes, despite the distance between their binding sites on linear DNA (*Bernard and Harley, 2010*; *Hou et al., 2017*).

The possibility of Sox-Bcat-Tcf ternary complexes raises the question of competitive vs cooperative DNA binding that, in turn, could influence transcriptional repression versus activation. For example, Sox17 might stabilize the formation of Bcat-Tcf-Ep300 complexes or destabilize the binding of co-repressors such as Tle and Hdac to promote transcription (e.g. *osr1*). To negatively regulate Bcat-Tcf activity (e.g. *dkk1*) Sox17 might destabilize Bcat-Tcf-Ep300-DNA or stabilize Tcf-Tle complexes. This idea is supported by work in mouse cortical neuron progenitors where Sox2 binds within 4–10 bp of Tcf/Lef sites and recruits Tle/Gro corepressors to repress Tcf/Bcatmediated *Ccnd1* expression (*Hagey and Muhr, 2014*). Future biochemical studies will need to determine if and how Sox-Bcat-Tcf complexes assemble on DNA.

### Broader implications

Although a growing number of non-Tcf Bcat-binding TFs have been reported, (*Gammons and Bienz, 2018*; *Notani et al., 2010*; *Trompouki et al., 2011*), how these impact the genomic specificity of Wnt-responsive transcription has not been well characterized and the idea of Tcf-independent Wnt transcription remains controversial (*Doumpas et al., 2019*; *Schuijers et al., 2014*). Our data suggest that in the gastrula endoderm, Sox17 might influence Bcat recruitment to chromatin to control cell-type-specific Wnt-responsive transcription, and that in some cases, this occurs independent of Tcfs. Given that there are 20 Sox TFs encoded in the vertebrate genome with most cell types expressing at least one, it is possible that Sox factors are widespread and previously unappreciated accessory effectors of Bcat that finetune Wnt-regulated transcription across many biological contexts.

## Materials and methods

### *Xenopus* embryo manipulations

*Xenopus laevis* and *Xenopus tropicalis* were purchased from Nasco or the *National Xenopus Resource* (RRID:SCR_013731) and experiments were performed according to CCHMC IACUC approved protocols. Embryos were staged according to *Nieuwkoop and Faber, 1967*. Microinjection and embryo culture were performed as previously described (*Stevens et al., 2017*). Antisense morpholino oligos (MO; GeneTools, LLC) were as follows:

Standard control MO: 5'CCTCTTACCTCAGTTACAATTTATA-3';
*Xenopus tropicalis* ß-*catenin* MO: 5' TTTCAACAGTTTCCAAAGAACCAGG 3';
*Xenopus tropicalis* sox17a MO: 5'-AGCCACCATCAGGGCTGCTCATGGT-3'
*Xenopus tropicalis* sox17b.1/2 MO: 5'-AGCCACCATCTGGGCTGCTCATGGT-3'
*Xenopus laevis* ß-*catenin* MO: 5'TTTCAACCGTTTCCAAAGAACCAGG-3' (*Heasman et al., 2000*);
*Xenopus laevis* sox17a MO: 5'-ATGATGAGGAGGGTTGGACAGGAGA-3' (*Clements et al., 2003*);

*Xenopus laevis* sox17b1/2 MO: 5'-TGATTCGGAGTGCTGTGGTGATTAG-3' (*Clements et al., 2003*);

Total amounts of MO injected into *X. tropicalis* embryos were as follows: 2 ng Bcat MO, 8 ng sox17 MO (4 ng each a+b1/2 MO), 8 ng standard control MO.

Synthetic mRNA for injections was generated using the Message Machine SP6 transcription kit (Thermo Fisher AM1340) using the following plasmids: pCS2+ human B-catenin S37A caBCAT (*Zorn et al., 1999*); pcDNA6-V5-mouse Sox17 (mSox17) and pcDNA6-V5-Sox17 M76A (*Sinner et al., 2007*); pCS2+ Tcf7l1:VP16 (formerly XTcf3:VP16) (*Darken and Wilson, 2001*).

## In situ hybridization

In situ hybridization of *Xenopus tropicalis* embryos was performed as described (*Sive, 2000*) with the following minor modifications. After overnight fixation at 4°C in MEMFA, embryos were washed (2 × 10 min) in MEMFA buffer without the formaldehyde and stored in 100% ethanol at −20°C. Embryos were serially rehydrated to PBS+0.01% Tween-20 and bisected through the dorsal-ventral axis in on 2% agarose-coated dishes using a fine razor blade (Electron Microscopy Services 72003–01), followed by Proteinase K (ThermoFisher AM2548) treatment at 1 µg/mL for 10 min. TEA/acetic anhydride steps were skipped on day 1. The RNAse A step was 0.25 µg/mL for 5 min on day 2; and finally the anti-DIG-alkaline phosphatase antibody (Sigma 11093274910) was used at a 1:6000 dilution in MAB buffer + 20% heat-inactivated lamb serum + 2% blocking reagent (Sigma 11096176001) on day2/3. Anti-sense DIG labeled in situ probes were generated using linearized cDNA templates with the 10X DIG RNA labeling mix (Sigma 11277073910) according to manufacturer's instructions. Details on plasmids used for probe synthesis are listed in *Supplementary file 7*.

## Antibodies and immunostaining

Affinity purified anti-*Xenopus* Sox17 rabbit polyclonal antibodies were generated by Bethyl labs to the Sox17a/b N-terminal, Sox17b C-terminal and Sox17a C-terminal with the peptides illustrated in *Figure 2—figure supplement 1*. Antibodies were validated by immunostaining, western blot, immunoprecipitation and ChIP. Immunoreactivity was not detected in Sox17-deficient tissue and specifically competed by addition of the target peptide to reactions. In preliminary ChIP-qPCR experiments, the pan-Sox17 and Sox17bC-terminal antibodies bound to the same loci, but the Sox17bC-terminal was more efficient and was used for ChIP-seq.

For *Xenopus* immunofluorescence, embryos were fixed in ice-cold 100 mM NaCl, 100 mM HEPES (pH7.5), 3.7% methanol-free paraformaldehyde for 1.5 hr at 4°C and then dehydrated directly into ice-cold Dent's post-fixative (80%Methanol/20% DMSO) and stored at −20°C. Embryos were serially rehydrated on ice into PBS+0.1% TritonX-100 (PBSTr) and bisected through the dorsal-ventral axis with a fine razor blade. Embryo halves were subjected to antigen retrieval in 1x R-Universal epitope recovery buffer (Electron Microscopy Services #62719–10) for 30 min at 55–60°C, washed 2 × 10 min in PBSTr, blocked for 2 hr in PBSTr + 10% normal donkey serum (Jackson Immunoresearch 017-000-001) + 0.2% DMSO at room temperature, and then incubated overnight at 4°C in this blocking solution + the following primary antibodies: chicken anti-GFP (Aves GPF-1020; diluted 1:1000), rabbit anti-Sox17a/b N-terminal (1:350), rabbit anti-B-catenin (Santa Cruz Biotechnology sc-7199; 1:500). After extensive washing in PBSTr, samples were incubated overnight at 4°C in PBSTr + secondary antibodies: donkey antiChicken 488, donkey anti-rabbit Cy3 (Jackson ImmunoResearch # 703-546-155, #711-166-152, both used at 1:1000 dilution).TO-PRO3 nuclear stain (ThermoFisher #T3605) was included in one 30 min PBSTr wash after secondary antibody incubation. Samples were extensively washed in PBSTr, dehydrated into 100% methanol, cleared and imaged in Murray's Clear (two parts benzyl benzoate, one part benzyl alcohol).

## Luciferase assays

Putative enhancers (wildtype as well as Sox and Tcf DNA-binding site mutants) were synthesized (supplementary sequence file S1) and cloned into the pGL4.23 *luc2*/miniP vector. (Promega E8411). *Xenopus laevis* embryos were used for luciferase assays because of the larger size embryos and slower developmental rate. Enhancer activity was assayed by co-injecting embryos with 5 pg of pRL-TKrenilla luciferase vector (Promega E2241), 50 pg of the pGL4.23 *luc2*/miniPenhancer:luciferase and combinations of the following MOS or mRNAs: ca-Bcat; 5 pg, mSox17; 25 pg, mSox17 M76A;

25 pg, Tcf3-VP16; 10 pg, *Xl sox17a+b1/2* MO; 10 ng each, *Xl bcat* MO; 10 ng. For each condition three tubes of 5 NF10.5 embryos were collected. Embryos were lysed in 100 µL of 100 mM TRIS-Cl pH7.5, centrifuged for 10 min at ~13,000 x *g* and then 25 µL of the clear supernatant lysate was used separately in Firefly (Biotium #30085–1) and Renilla (Biotium 300821) luciferase assays according to the manufacturer's instructions. The average relative Luciferase activity was normalized Renilla levels for each sample and statistical significance was determined by pairwise Student T-test. Each luciferase experiment was repeated two to four times with similar results and a representative is shown.

## RNA-sequencing

### RNA isolation and library preparation

Approximately 200 *X. tropicalis* embryos were injected with 8 ng of either control-MO or Sox17-MO, or 2 ng of control-MO and Bcat-MO and 10 sibling embryos were collected at every time point from blastula to late gastrula in biological duplicate. Total RNA was extracted with the Nucleo-spin RNA kit (Machery-Nagel). RNA-seq libraries were generated using Smart-seq2 cDNA synthesis followed by tagmentation, quality-tested using an Agilent Bioanalyzer 2100, quantified using KAPA qPCR and sequenced using Illumina sequencers at the UC Irvine Genomics High Throughput Facility.

### RNA-Seq analysis

Raw reads were quality checked using FastQC and adapters/low-quality reads were removed with Trimmomatic (*Bolger et al., 2014*). Reads were mapped to the *Xenopus tropicalis* genome version 9.0 (*Hellsten et al., 2010*) with bowtie2, quantified using RSEM (*Li and Dewey, 2011*) and reported in transcripts per million (TPM). *X. tropicalis* v9.0 gene name annotations were obtained from www.Xenbase.org; RRID:SCR_003280 (*Karimi et al., 2018*). To visualize the data, aligned bam files generated as above were converted to bigwig format using deeptools v3.4.1 and visualized on IGV v2.4.1.3 (*Thorvaldsdóttir et al., 2013*). Differentially expressed genes ($\geq 2$ fold change and FDR $\leq$ 5%) were identified by a pairwise comparison between control-MO versus Sox17-MO and control-MO versus Bcat-MO sibling embryos at each stage using RUVSeq (R package) (*Risso et al., 2014*). Gene ontology enrichment analysis was performed using the Gene Ontology Consortium Online Resource (http://geneontology.org) and default parameters.

To assess the spatial expression of transcripts, we examined the RNA-seq TPM expression values of dissected stage 10.5 embryo published in GEO GSE81458 (*Blitz et al., 2017*). Enriched expression in a specific tissue was defined as the normalized expression being >1.5 x of the mean expression across dissected tissues. Statistical significance of enriched expression pattern in groups of genes was determined by chi-square test performed in R package MASS.

## Chromatin immunoprecipitation

### ChIP-Seq

Large-scale Sox17 ChIP was carried out as previously described (*Charney et al., 2017a*) using 2000 NF10.5 *X. tropicalis* embryos and 30 µg rabbit anti-Sox17b-Cterminal antibody. ChIP-seq libraries were generated using Nextflex ChIP-seq kit (Bio Scientific), analyzed using an Agilent Bioanalyzer 2100, quantified using KAPA qPCR and sequenced using Illumina instruments at the UC Irvine Genomics High Throughput Facility.

### ChIP-qPCR

was carried out largely as described in *Charney et al., 2017a*; *Hontelez et al., 2015* with minor modifications. Briefly, 50–200 NF 10.5 *Xenopus tropicalis* embryos were fixed in 1% formaldehyde in 0.1x MBS at room temperature for 30 min. Sonication was carried out in a Diagenode Bioruptor Pico for 30 cycles of 30 s ON, 30 s OFF. Chromatin extracts were incubated with 10 µg rabbit anti-Sox17-Cterminal antibody (affinity purified by Bethyl labs, *Figure 2—figure supplement 1*) or 2.5 µg rabbit anti-β-catenin (Life Technologies, 712700) overnight at 4°C. After washes and elution, DNA was then purified using a MinElute PCR Purification kit and QIAQuick purification columns and eluted in 20 µl TE.

## ChIP-reChIP

Before the first ChIP, the anti-β-catenin antibody was crosslinked to Protein G Dynabeads. ChIP assay was then carried out on extracts from 500 embryos as described above. At the end of the first ChIP, DNA was eluted with elution buffer supplemented with 10 mM DTT. The eluate was then diluted in 10 volumes of wash buffer (50 mM Tris-HCl pH 8.0, 100 mM NaCl, 2 mM EDTA, 1% NP-40) supplemented with 1x Protease Inhibitor Cocktail and 1 mM DTT. The 2nd ChIP was then carried out as described above.

## ChIP-Seq analysis

Raw reads were assessed by FastQC and trimmed with Trimmomatic and aligned to *Xenopus tropicalis* genome (*Hellsten et al., 2010*) version 9.0 using bowtie2 v2.3.4.7 (*Langmead and Salzberg, 2012*). Duplicate and multi-mapped reads were removed by Picard (http://broadinstitute.github.io/picard/) and samtools (*Li et al., 2009*). Peaks were called with MACS2 (*Zhang et al., 2008*) against NF10.5 input DNA with default options. HOMER v4.1.0 (*Heinz et al., 2010*) was used to annotate ChIP-seq peaks (intron, intergenic, exon and promoter) and to associate peaks with genes using the nearest TSS function. To identify statistically significant ChIP-seq peaks, we performed Irreproducible discovery rate (IDR) analysis per ENCODE Best Practices guidelines (*Landt et al., 2012*) on pseudoreplicate from pooled Sox17 ChIP-seq biological replicates and pseudoreplicate of the single Bcat ChIP-seq dataset from GEO GSE72657 (*Nakamura et al., 2016*) with an IDR threshold of 0.01. To visualize the data, aligned bam files generated as above were converted to bigwig format using deeptools v3.4.1 and visualized on IGV v2.4.1.3 (*Thorvaldsdóttir et al., 2013*). Significance of intersected ChIP-seq and RNA-seq datasets was assessed using hyper geometric test (HGT).

## Motif analysis

De-novo motif analysis was performed with HOMER v4.1.0 (*Heinz et al., 2010*). For a comprehensive Sox and Tcf motif analysis, we divided our datasets into three groups: (a) Peaks bound and regulated by Sox17 only, (b) Peaks bound and regulated by Bcat Only and (c) Peaks that were cobound and coregulated by Sox17 and Bcat. DNA sequences of 150 bp centered on the peaks summits were extracted and motif analysis was carried out as follows: (1) To find which peaks have instances of Sox17 and Tcf motifs, HOMER function annotatePeaks.pl was used with the parameters: -size 150 m motif sox17.motif tcf7.motif lef1.motif tcf3.motif tcf7l2.motif. (2) To perform a more exhaustive motif search the enhancers were also scanned with a library of position weight matrices (PWMs) containing all experimentally defined Sox17, Tcf7, Tcf7l1 and Lef1 motifs $\geq$ 70% threshold from the CIS-BP database (*Lambert et al., 2019*). The HOMER and CIS-BP results were merged to generate BED files and Bedtools v2.27.1 (*Quinlan and Hall, 2010*) intersect function was then used to extract coordinates of 'Sox only' 'Tcf only' and 'Sox and Tcf overlapping' categories. Fisher's exact tests were used to determine the statistical significance of Sox17 or Tcf motif frequency in each set of peaks compared to random background DNA generated by randomly scrambling each wild-type test sequence (maintaining dinucleotide composition) ten times to generate randomly shuffled sequences that had the same GC content.

## Peak density heatmaps

From the Sox17 and Bcat peaks called by MACS2, Bedtools intersect function was used to stratify the peaks into 'Sox bound only', 'Bcat bound only' and 'Sox and Bcat overlapping peaks'. For each category of peaks, the peak summit was extracted from the TF summit file generated by MACS2 and 2000 bp were added to both sides of the summit using awk. Bedtools computematrix was used to generate per genome regions. The generated matrix was then used with bedtools plotHeatmap to generate heatmaps and signal density plots.

## Analysis of public datasets

The following datasets were downloaded from GEO: Embryo dissection RNA-seq; GSE81458 (*Blitz et al., 2017*), embryonic stage series RNA-seq; GSE65785; (*Owens et al., 2016*), Bcat ChIP-seq; GSE72657 (*Nakamura et al., 2016*), RNA pol II and Foxa ChIP-seq; GSE85273 (*Charney et al., 2017a*), H3K27Ac ChIP-seq; GSE56000 (*Gupta et al., 2014*), H3K4me1 and Ep300 ChIP-seq; GSE67974 (*Hontelez et al., 2015*), Smad1, Smad2, Sox3 and VegT ChIP-seq; GSE113186

(*Gentsch et al., 2019*). Raw data was downloaded from GEO and aligned to the *Xenopus tropicalis* genome version 9.0, and processed to generate bigwig files and heatmaps as described above.

## Acknowledgements

This work was supported by HD073179 to AMZ and KWYC and by NIH P30 DK078392 gene expression core of the Digestive Diseases Research Core Center in Cincinnati. We thank the University of California (UC) Irvine Genomic High-Throughput Facility Shared Resource of the

Cancer Center Support Grant (P30CA-062203) and the UC Riverside Institute for Integrative Genome Biology for sequencing support. We are grateful to members of the Zorn and Wells labs and the Endoderm Club for helpful discussions. We also thank Xenbase.org (RRID:SCR_003280) and the National *Xenopus* Resource (RRID:SCR_013731) for critical community resources.

## Additional information

### Funding

| Funder | Grant reference number | Author |
|---|---|---|
| Eunice Kennedy Shriver National Institute of Child Health and Human Development | HD073179 | Ken WY Cho<br>Aaron M Zorn |
| National Institute of Diabetes and Digestive and Kidney Diseases | P30DK078392 | Aaron M Zorn |
| Eunice Kennedy Shriver National Institute of Child Health and Human Development | P01HD093363 | Aaron M Zorn |

The funders had no role in study design, data collection and interpretation, or the decision to submit the work for publication.

### Author contributions

Shreyasi Mukherjee, Conceptualization, Data curation, Software, Formal analysis, Validation, Investigation, Visualization, Methodology, Writing - original draft; Praneet Chaturvedi, Software, Formal analysis, Writing - review and editing; Scott A Rankin, Conceptualization, Formal analysis, Validation, Investigation, Methodology, Writing - review and editing; Margaret B Fish, Marcin Wlizla, Investigation, Methodology; Kitt D Paraiso, Software, Visualization, Writing - review and editing; Melissa MacDonald, Validation; Xiaoting Chen, Software, Formal analysis; Matthew T Weirauch, Software, Supervision, Methodology; Ira L Blitz, Conceptualization, Formal analysis, Supervision, Validation, Investigation, Methodology, Writing - review and editing; Ken WY Cho, Conceptualization, Supervision, Funding acquisition, Project administration, Writing - review and editing; Aaron M Zorn, Conceptualization, Resources, Data curation, Formal analysis, Supervision, Funding acquisition, Visualization, Writing - original draft, Project administration, Writing - review and editing

### Author ORCIDs

Aaron M Zorn (ID) https://orcid.org/0000-0003-3217-3590

### Ethics

Animal experimentation: This study was performed in strict accordance with the recommendations in the Guide for the Care and Use of Laboratory Animals of the National Institutes of Health. All of the animals were handled according to approved institutional animal care and use committee (IACUC) protocol (#AU2_IACUC2016-0059) of the Cincinnati Children's Hospital Medical Center.

### Decision letter and Author response

Decision letter https://doi.org/10.7554/eLife.58029.sa1
Author response https://doi.org/10.7554/eLife.58029.sa2

# Additional files

## Supplementary files

- Supplementary file 1. Sox17-regulated transcripts. An excel spread sheet of differentially expressed genes from RNA-sequencing of control and Sox17-MO embryos across stages NF 9–12 and associated metadata annotations.

- Supplementary file 2. Sox17 ChIP-seq peaks. Coordinates of Sox17-bound peaks at NF10.5 with their gene annotations.

- Supplementary file 3. β-Catenin-regulated transcripts. An excel spread sheet of differentially expressed genes in RNA-sequencing of control and Bcat-MO embryos across stages NF 9–12 and associated metadata annotations.

- Supplementary file 4. β-Catenin ChIP-seq peaks. Coordinates of Bcat-bound peaks at NF10.5 with their gene annotations. Primary data from GSE72657 (*Nakamura et al., 2016*).

- Supplementary file 5. Sox17- β-catenin cobound chromatin. Coordinates of 191 Sox17 and Bcat cobound/coregulated enhancers and the frequency of Sox and Tcf DNA-binding sites per peak.

- Supplementary file 6. Sox17- β-catenin cobound and regulated enhancers. Sequence of wild-type and Sox/Tcf mutated enhancers used for luciferase assays.

- Supplementary file 7. Plasmids and primers. An excel file of information of the cDNA plasmid used for in situ probe synthesis (tab-1) and primers for used for ChIP-qPCR (tab-2).

- Transparent reporting form

## Data availability

The RNA-seq and ChIP-seq data generated by this study have been deposited in the NCBI Gene Expression Omnibus (GEO) under accession GSE148726.

The following dataset was generated:

| Author(s) | Year | Dataset title | Dataset URL | Database and Identifier |
|---|---|---|---|---|
| Paraiso KD, Zorn AM, Mukherjee S | 2020 | Sox17 and $\beta$-catenin co-occupy Wnt-responsive enhancers to govern the endodermal gene regulatory network | https://www.ncbi.nlm. nih.gov/geo/query/acc. cgi?acc=GSE148726 | NCBI Gene Expression Omnibus, GSE148726 |

The following previously published datasets were used:

| Author(s) | Year | Dataset title | Dataset URL | Database and Identifier |
|---|---|---|---|---|
| Blitz IL, Paraiso KD | 2016 | Regional expression of X. tropicalis transcription factors in early gastrula embryos | https://www.ncbi.nlm. nih.gov/geo/query/acc. cgi?acc=GSE81458 | NCBI Gene Expression Omnibus, GSE81458 |
| Owens ND, Blitz IL, Lane MA, Patrushev I, Overton JD, Gilchrist MJ, Cho KW, Khokha MK | 2016 | Measuring Absolute RNA Copy Numbers at High Temporal Resolution Reveals Transcriptome Kinetics in Development | https://www.ncbi.nlm. nih.gov/geo/query/acc. cgi?acc=GSE65785 | NCBI Gene Expression Omnibus, GSE65785 |
| Nakamura Y, Alves E, Hoppler S | 2016 | Tissue- and stage-specific cellular context regulates Wnt target gene expression subsequent to $\beta$-catenin recruitment | https://www.ncbi.nlm. nih.gov/geo/query/acc. cgi?acc=GSE72657 | NCBI Gene Expression Omnibus, GSE72657 |
| Charney RM, Cho KW | 2017 | Foxh1 marks the embryonic genome prior to the activation of the mesendoderm gene regulatory program | https://www.ncbi.nlm. nih.gov/geo/query/acc. cgi?acc=GSE85273 | NCBI Gene Expression Omnibus, GSE85273 |
| Gupta R, Baker JC | 2014 | Enhancer chromatin signatures predict Smad2/3 binding in *Xenopus* | https://www.ncbi.nlm. nih.gov/geo/query/acc. cgi?acc=GSE56000 | NCBI Gene Expression Omnibus, GSE56000 |

| Hontelez S, Veenstra GC | 2015 | Embryonic transcription is controlled by maternally defined chromatin state | https://www.ncbi.nlm.nih.gov/geo/query/acc.cgi?acc=GSE67974 | NCBI Gene Expression Omnibus, GSE67974 |
|---|---|---|---|---|
| Gentsch GE, Smith JC | 2019 | Maternal pluripotency factors initiate extensive chromatin remodelling to predefine first response to inductive signals | https://www.ncbi.nlm.nih.gov/geo/query/acc.cgi?acc=GSE113186 | NCBI Gene Expression Omnibus, GSE113186 |
| Meissner A | 2015 | Transcription factor binding dynamics during human ES cell differentiation | https://www.ncbi.nlm.nih.gov/geo/query/acc.cgi?acc=GSE61475 | NCBI Gene Expression Omnibus, GSE61475 |

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
