## [Decision Letter]

**Acceptance summary:**

This is an interesting and important contribution that addresses the control of endodermal gene expression through Sox17 and Wnt β-catenin. The experiments demonstrate that Sox17 acts as an endoderm-specific modifiers of Wnt-signaling responses. The study also fills an important gap in our understanding of early cell fate specification in the embryo by identifying the direct targets of Sox17 in the endoderm.

**Decision letter after peer review:**

Thank you for submitting your article "Sox17 and β-catenin co-occupy Wnt-responsive enhancers to govern the endodermal gene regulatory network" for consideration by *eLife*. Your article has been reviewed by three peer reviewers, and the evaluation has been overseen by a Reviewing Editor and Edward Morrisey as the Senior Editor. The reviewers have opted to remain anonymous.

The reviewers have discussed the reviews with one another and the Reviewing Editor has drafted this decision to help you prepare a revised submission.

As the editors have judged that your manuscript is of interest, but as described below, that additional experiments may be required before it is published, we would like to draw your attention to changes in our revision policy that we have made in response to COVID-19 (https://elifesciences.org/articles/57162). First, because many researchers have temporarily lost access to the labs, we will give authors as much time as they need to submit revised manuscripts. We are also offering, if you choose, to post the manuscript to bioRxiv (if it is not already there) along with this decision letter and a formal designation that the manuscript is "in revision at *eLife*". Please let us know if you would like to pursue this option. (If your work is more suitable for medRxiv, you will need to post the preprint yourself, as the mechanisms for us to do so are still in development.)

Summary:

This paper by Mukherjee et al. addresses functional interactions of Sox17 and Wnt signaling in endodermal gene regulation. The results reveal that Sox17 factors act as endoderm-specific modifiers of Wnt-signaling responses, providing strong evidence for a TCF independent role for β-catenin/Sox17 in promoting endodermal and suppressing mesodermal/ectodermal gene expression. The reviewers felt that this interesting study addresses important questions and is well-executed, but that there are instances where the conclusions seem overgeneralized, and that some additional analysis is needed before the article would be appropriate for publication in *eLife*.

Essential revisions:

1) Interpreting the Sox17 and β-catenin knock down results requires assessment of the efficiency knock downs – is occupancy at high affinity sites being decreased or only low affinity? Ideally this would be shown by ChIP following knockdown.

2) The generalized finding that "Sox17 and β-catenin co-occupy a subset of enhancers to synergistically stimulate transcription in the absence of Tcfs" is not fully supported by the data presented. Although a subset of enhancers lack predicted Tcf binding motifs, functional data is shown only for *six1*, and only by gain-of function experiments. Data showing that other enhancers behave like *six1*, and possibly also using a Tcf7l1-EnR fusion, are needed to support this conclusion. Absent such data, conclusions should relate only to Six1 regulation.

3) Please add discussion of the dynamics of Sox17 binding/function. Could this explain differences with reported enrichment in other studies such as Nakamura et al. and Kjolby et al.? Those differences should be discussed.

4) It might be useful to extend the analysis to look at collaborating transcription factors (e.g. Tbx, Pou, etc) by cross-referencing with published data sets (for example such as Gentsch et al.).

5) In the first section of the Results, the text should reflect the possibility that some of the observed transcriptional effects after Sox17 knockdown are caused by indirect mechanisms. Also, please avoid using the non-specific term "regulates", and instead use "inhibits/increases".

6) Motif frequencies in peaks cannot be compared directly between motifs or different versions of the motifs, as seems to have been done in Figure 4—figure supplement 2A because the baseline for motif occurrence (in random genomic sequences) is highly motif-specific.

7) Please indicate the application of multiple testing correction (adjusted p-value, or FDR) where applicable (Materials and methods, Results; GO analysis). Also indicate p-values for all enrichments and intersections (Figures 2D, Figure 2—figure supplement 2D, Figure 3—figure supplement 2B, Figure 4A).

8) Including a summary of the temporal role of Sox17 in context of other signals and transcription factors, including Nodal/Smad2, BMP, VegT would be useful.

[Editors' note: further revisions were suggested prior to acceptance, as described below.]

Thank you for resubmitting your work entitled "Sox17 and β-catenin co-occupy Wnt-responsive enhancers to govern the endoderm gene regulatory network" for further consideration by *eLife*. Your revised article has been evaluated by Edward Morrisey (Senior Editor) and a Reviewing Editor.

The manuscript has been greatly improved but there is one remaining issue that should be addressed before acceptance. This concerns the motif analysis in Figure 4—figure supplement 2. Examining background frequencies via scrambling of peak sequences while maintaining di-nucleotide frequencies has strengthened this analysis. However, ranking the motifs by hypergeometric p-value (provided by HOMER) is perhaps not most informative way to represent these data. We suggest instead plotting the 2log fold enrichment over background, in combination with a statistical test (effect size and statistical rigor). Also, it is felt that a discussion of background frequencies is needed where the co-occurring motifs are discussed and Figure 4—figure supplement 2D should be revised by calculating for background sequences.

---

## [Author Response]

Essential revisions:1) Interpreting the Sox17 and β-catenin knock down results requires assessment of the efficiency knock downs – is occupancy at high affinity sites being decreased or only low affinity? Ideally this would be shown by ChIP following knockdown.

We agree that this is an important point. We have now performed ChIP-qPCR experiments on control, Sox17-MO and β-catenin-MO embryos as requested. The new data added to Figure 2—figure supplement 1D-E shows a dramatic loss of ChIP-qPCR signal at all loci tested, down to negative control background levels in most cases. This data supports our immunostaining (Figure 1A-C, and Figure 3—figure supplement 1A) showing a complete loss of nuclear Sox17 and β-catenin in knockdown embryos. Together from all the data we estimate that the knockdown efficiency is greater than 90%.

The issue of high-affinity versus low-affinity site is very interesting and we understand the reviewers point that, with a partial knockdown, high-affinity enhancers might still be active. Since our knockdown is very efficient, we don't expect this will play much of a role, but we cannot totally rule it out. This is difficult to test since in vivo high-affinity and low-affinity sites for Sox17 (or Tcfs) have not yet been characterized in gastrula embryos. In an attempt to identify high-affinity vs. low-affinity Sox17 binding sites, we analyzed published protein binding microarray data from the CIS-BP database where recombinant DNA-binding domains of hundreds of TFs were challenged to all possible combinations of DNA 8-mers (Badis et al., 2009; Lambert et al., 2019). Per the original criteria set out by Badis et al., we classified 8-mer sequences that had an enrichment (E)-score > 0.45 for Sox17 binding as “high affinity” sequences and those with an E-score < 0.2 as “low affinity” sequences. To assess if any of these sequences were present in our in vivo ChIP-seq data, we scanned Sox17 bound peaks for the presence of potential Sox binding sites. We then selected candidate enhancers for ChIP-qPCR in Figure 2—figure supplement 1D to include some that were predicted to contain only high-affinity Sox-sites (*foxi2* and *snai2)* or low-affinity Sox17 sites (*nrp1* and *lmbx1).* ChIP-qPCR shows that Sox17 occupancy dramatically decreased by several orders of magnitude at all four enhancers, after Sox17-MO knockdown. However, there was still some signal above background at *foxi2*. We prefer to not make any conclusions about this result, because these experiments lack the rigorous quantitative measurement of DNA-binding affinity needed for such claims, and this is not really the point of the current study. In the future we plan to systematically examine the impact of DNA-affinity which is complicated because Sox monomers bind to DNA with low sequence specificity and low affinity and are thought to gain specificity by other interaction with co-occupying transcription factors (Lefebvre et al., 2007; Hou et al., 2017). We have added this point to the Discussion.

2) The generalized finding that "Sox17 and β-catenin co-occupy a subset of enhancers to synergistically stimulate transcription in the absence of Tcfs" is not fully supported by the data presented. Although a subset of enhancers lack predicted Tcf binding motifs, functional data is shown only for six1, and only by gain-of function experiments. Data showing that other enhancers behave like six1, and possibly also using a Tcf7l1-EnR fusion, are needed to support this conclusion. Absent such data, conclusions should relate only to Six1 regulation.

We have now extended our analysis to include two additional enhancers: 1) a +7kb *osr1* and 2) +3kb *igf3* enhancer that shared similar characteristics to *six1;* (they’re expressed in the endoderm, activated by Sox17 and β-catenin and have no evidence of TCF binding sites). The new Figure 6—figure supplement 1 shows that Sox17 and β-catenin cooperatively activate both enhancers, but neither was activated by Tcf7l1-VP16 (similar to *six1* but unlike the *dkk1* and *lhx5* enhancers which contain Tcf-sites and are activated by Tcf7l1-VP16). These new data further support our conclusion that “Sox17 and b-catenin co-occupy a subset of enhancers to synergistically stimulate transcription *independently* of Tcfs”. We have not tried the Tcf7l1-EnR fusion. This reagent is problematic as it is well known to dominantly block all maternal Wnt/β-catenin-targets including *nodal*. Thus the embryos arrest without ever initiating endoderm fate or *sox17* expression, precluding the analysis. Acknowledging that we don’t have a Tcf loss of function, we have now tempered our conclusions to say “apparently independent of Tcfs” in the Abstract and throughout the paper. We also point out in the paper that the majority of Sox17-β-catenin cobound enhancers do have Tcf sites (even those that are activated). In the future we plan to fully characterize the role of Tcfs at these enhancers. We emphasize the enhancers without detectable Tcf-regulation since this is the most novel aspect of the work.

3) Please add discussion of the dynamics of Sox17 binding/function. Could this explain differences with reported enrichment in other studies such as Nakamura et al. and Kjolby et al.? Those differences should be discussed.

As requested, we have added a section in the Discussion on the temporal dynamics of Sox17 binding/function. Moreover, we have added a new panel to Figure 3—figure supplement 2 comparing our motif enrichment analysis to that of Kjolby et al., 2017 and Gentsch et al., 2019. We should point out that we use the published β-catenin ChIP-seq data reported in Nakamura et al., 2016 (stage 10.5 *Xenopustropicalis* ChIP-seq of endogenous β-catenin). Although our processing pipeline is a little bit different than theirs, like them we find >11,000 IDR peaks and our motif analysis shows Sox (2^nd^ ranked) being more enriched than Tcfs (5^th^ ranked). This is almost exactly what Nakamura et. al reported. Using the same pipeline and motif analysis of β-catenin bound peaks of the Gentsch et al., 2019 dataset (β-catenin ChIP-seq at stage 10+), we find a similar ranking of Sox (3^rd^ ranked) and Tcf (6^th^ ranked) motifs. In the case of Kjolby et al., they performed ChIP-seq on slightly later stage NF11.5 *Xenopus laevis* using an overexpressed 3xFlag-tagged β-catenin reporting only 855 reproducible peaks. HOMER de novo motif analysis of the Kjolby data shows Lef1 as the 1^st^ top motif and Sox as the 3^rd^ ranked (Figure 3—figure supplement 2C). So essentially there is not much difference in the motif analysis reported in these different studies. In both cases, Sox and Tcf motifs are among the top 5 enriched in β-catenin bound peaks, these other studies just did not follow up on the high rank of Sox motifs. The minor differences in motif calling between papers could be due to many things: the number of peaks in the analysis; >10,000 peaks vs. 855 peaks, developmental timing; 10 vs. 11.5, species; *X. tropicalis* vs. *X. laevis,* overexpression vs. endogenous, antibody; β-catenin vs. Flag or different lab ChIP protocols. Since we have no way of knowing the source of variation, we prefer not to speculate in the discussion.

4) It might be useful to extend the analysis to look at collaborating transcription factors (e.g. Tbx, Pou, etc) by cross-referencing with published data sets (for example such as Gentsch et al.).

Per the reviewers’ suggestion, we extended our analysis to all other publicly available datasets of endodermal TFs (Charney et. al, 2017, Paraiso et al., 2019, Gentsch et. al, 2019). This new analysis presented in Figure 4—figure supplement 1, shows that Sox17 and β-catenin co-occupied peaks were indeed enriched for binding of other endodermal TFs including Foxa4, Smad1, Smad2, Sox3 and VegT. We have described this in the Results and Discussion sections explaining how it is likely that combinatorial activity of multiple lineage-specific TFs may cooperate with Sox17/β-catenin on super enhancers to regulate the endoderm GRN.

5) In the first section of the Results, the text should reflect the possibility that some of the observed transcriptional effects after Sox17 knockdown are caused by indirect mechanisms. Also, please avoid using the non-specific term "regulates", and instead use "inhibits/increases"

We have now updated the text as requested and explicitly indicated that changes in transcript level after Sox17-depletion could be indirect.

6) Motif frequencies in peaks cannot be compared directly between motifs or different versions of the motifs, as seems to have been done in Figure 4—figure supplement 2A because the baseline for motif occurrence (in random genomic sequences) is highly motif-specific.

We thank the reviewers for bringing this to our attention. We agree that since the occurrence of different motifs or version of motifs in random genomic sequences is variable, making comparisons not really possible. In the revisions, we have modified our analysis to not directly compare between Sox and Tcf motif frequencies, but to ask whether the frequency of Sox or Tcf motifs in a given group or peaks is significant over random. In addition we used a library of all predicted Sox or Tcf motifs rather than a specific motif.

To assesses motif frequency in different categories of peaks we used a library of all known Sox and Tcf motifs combining HOMER and an exhaustive search of the CIS-BP database containing all experimentally defined sites from protein-DNA binding microarrays (Badis et al., 2009; Lambert et al., 2019). Combining these (all HOMER instance were also in CIS-BP) we generated a library of position weight matrices (PWMs) of all potential Sox17, Tcf7l1, Tcf7l2, Tcf7, Lef1 motifs and scanned peak scoring the number of PWMs present above a 70% threshold. A list of all the motifs from the CIS-BP database that we’ve included in our analysis is show in Author response table 1. We had not explained this properly in the original version of the paper and have now updated the Materials and methods.

Author response table 1

Then for each wild-type test peak we generated a random “shuffled” version where the wild-type peak sequences were randomly scrambled ten times maintaining dinucleotide frequencies. These represented random genomic sequences with the same GC content as the test case. For each iteration, we determine the frequency of the Sox and Tcf motifs in the “wild-type” and “shuffled” peaks. We then perform enrichment analysis using Fisher’s exact test to determine whether the frequency of Sox or Tcf motifs in a given group of “wild-type” peaks was enriched over the random “shuffled” sequences. Our results reveal Sox17 motifs are more frequently enriched in Sox17 bound and Sox17/β-catenin bound peaks over random, while Tcf motifs are not enriched over random in any of the categories of sequences. We now incorporate this new analysis in the revised results and new Figure 4—figure supplement 2. In the future we will explore the biological impact of number and frequency of motifs in enhancers, but for the current paper this is a very minor observation, from which we do not make any major conclusions.

7) Please indicate the application of multiple testing correction (adjusted p-value, or FDR) where applicable (Materials and methods, Results; GO analysis). Also indicate p-values for all enrichments and intersections (Figures 2D, Figure 2—figure supplement 2D, Figure 5—figure supplement 2B, Figure 4A).

We have now updated the figures, legends and Materials and methods to indicate the statistical tests used and p-values for all enrichments and intersections.

8) Including a summary of the temporal role of Sox17 in context of other signals and transcription factors, including Nodal/Smad2, BMP, VegT would be useful.

We have now incorporated the requested summary in the Discussion.

[Editors' note: further revisions were suggested prior to acceptance, as described below.]

The manuscript has been greatly improved but there is one remaining issue that should be addressed before acceptance. This concerns the motif analysis in Figure 4—figure supplement 2. Examining background frequencies via scrambling of peak sequences while maintaining di-nucleotide frequencies has strengthened this analysis. However, ranking the motifs by hypergeometric p-value (provided by HOMER) is perhaps not most informative way to represent these data. We suggest instead plotting the 2log fold enrichment over background, in combination with a statistical test (effect size and statistical rigor).

As requested, we added a new panel Figure 4—figure supplement 2D plotting the log2 fold enrichment of Sox and Tcf site frequency for each class of peak compared to random shuffled sequence with a Fisher Exact test to show the enrichment and statistical significance. We agree that some readers may find this way of viewing the data easier. Other naïve test readers preferred the original display and so we kept that too.

Also, it is felt that a discussion of background frequencies is needed where the co-occurring motifs are discussed and Figure 4—figure supplement 2D should be revised by calculating for background sequences.

As requested, we modified this panel (now Figure 4—figure supplement 2E) to also show random genomic background sequence. A Kolmogorov-Smirnov test comparing the motif distribution of each group to background, showed that the “Sox17 and β-catenin” and “Sox17 Only” motif distribution was significantly different to random background. We made minor modifications to the text to reflect this.

We want to stress that the motif frequency and co-occurrence is very minor part of the paper. The only points we want to make are (1) that most peaks contain both Sox and Tcf sites while some have just Sox sites and (2) that when Sox motifs are present there are usually multiple sites. We did not make any assertions based on the statistics. In many respects we could remove Figure 4—figure supplement 2 completely and it would not change any of our conclusions. We just included these observations as the reader might be interested.